# Adam-family Methods with Decoupled Weight Decay in Deep Learning

**Kuangyu Ding**  *kuangyud@u.nus.edu*
*Edwardson School of Industrial Engineering*
*Purdue University*

**Nachuan Xiao**  *xncxy@cuhk.edu.cn*
*School of Data Science*
*The Chinese University of Hong Kong, Shenzhen*

**Kim-Chuan Toh**  *mattohkc@nus.edu.sg*
*Department of Mathematics*
*and Institute of Operations Research and Analytics*
*National University of Singapore*

**Reviewed on OpenReview:** *https://openreview.net/forum?id=xVEHiAZ7uR*

## Abstract

In this paper, we investigate the convergence properties of a wide class of Adam-family methods for minimizing quadratically regularized nonsmooth nonconvex optimization problems, especially in the context of training nonsmooth neural networks with weight decay. Motivated by AdamW, we propose a novel framework for Adam-family methods with decoupled weight decay. Within our framework, the estimators for the first-order and second-order moments of stochastic subgradients are updated independently of the weight decay term. Under mild assumptions and with non-diminishing stepsizes for updating the primary optimization variables, we establish the convergence properties of our proposed framework. In addition, we show that our proposed framework encompasses a wide variety of well-known Adam-family methods, hence offering convergence guarantees for these methods in the training of nonsmooth neural networks. More importantly, compared to the existing results on the choices of the parameters for the moment terms in Adam, we show that our proposed framework provides more flexibility for these parameters. As a practical application of our proposed framework, we propose a novel Adam-family method named Adam with Decoupled Weight Decay (AdamD), and establish its convergence properties under mild conditions. Numerical experiments demonstrate that AdamD outperforms Adam and is comparable to AdamW, in the aspects of both generalization performance and efficiency.

## 1 Introduction

We consider the following unconstrained optimization problem (UOP):

$$\min_{x \in \mathbb{R}^n} \quad g(x) := f(x) + \frac{\sigma}{2} \|x\|^2, \tag{UOP}$$

where the function $f : \mathbb{R}^n \to \mathbb{R}$ is assumed to be locally Lipschitz continuous and possibly nonsmooth over $\mathbb{R}^n$. Moreover, the constant $\sigma > 0$ is the penalty parameter for the quadratic regularization term. Such a regularization term is also known as the weight decay term, which is widely employed to enhance the generalization performance in training neural networks (Bos & Chug, 1996; Krogh & Hertz, 1991).

Stochastic Gradient Descent (SGD) is one of the most fundamental methods for solving (UOP). In SGD, all coordinates of the variable $x$ are updated using the same stepsize (i.e., learning rate). To accelerate SGD,

(Kingma & Ba, 2015) develops the widely used Adam method, which adjusts coordinate-wise stepsizes based on first-order and second-order moments of the stochastic gradients. Due to its high efficiency in training neural networks, Adam has become one of the most popular choices for various neural network optimization tasks.

Motivated by Adam, numerous efficient Adam-family methods have been developed, such as AdaBelief (Zhuang et al., 2020), AMSGrad (Reddi et al., 2018), Yogi (Zaheer et al., 2018), etc. From a theoretical perspective, the majority of existing works (Barakat & Bianchi, 2021; Défossez et al., 2022; Guo et al., 2021; Shi et al., 2021; Wang et al., 2022; Zaheer et al., 2018; Zhang et al., 2022; Zhuang et al., 2022; Zou et al., 2019) establish convergence properties for these Adam-family methods, based on the assumption that $f$ is continuously differentiable over $\mathbb{R}^n$. However, as emphasized in (Bolte et al., 2021; Bolte & Pauwels, 2021; Bolte et al., 2022b), nonsmooth activation functions, including ReLU and leaky ReLU, are popular choices in building neural networks. For any neural network built from these nonsmooth activation functions, its loss function is usually nonsmooth and lacks Clarke regularity (e.g., differentiability, weak convexity, etc.). Consequently, these existing works are unable to provide convergence guarantees for their analyzed methods in the training of nonsmooth neural networks.

## 1.1 Existing works on training nonsmooth neural networks

In nonsmooth optimization, it has been demonstrated in (Daniilidis & Drusvyatskiy, 2020) that a general Lipschitz continuous function $f$ can exhibit highly pathological properties, leading to the failure of subgradient descent method to find a critical point of $f$. Moreover, the chain rule may fail for the Clarke subdifferential (Clarke, 1990) of the loss function of a nonsmooth neural network. Specifically, when we differentiate the loss function of a nonsmooth neural network using automatic differentiation (AD) algorithms, the outputs may not be contained in the Clarke subdifferential of $f$ (Bolte & Pauwels, 2020).

Consequently, most of the existing works restrict their analysis to the class of *path-differentiable* functions (Bolte & Pauwels, 2021, Definition 3). For any path-differentiable function $f$, there exists a graph-closed set-valued mapping $\mathcal{D}_f$, called *conservative field* for $f$, such that for any absolutely continuous mapping $\gamma : [0, \infty) \to \mathbb{R}^n$, it holds that $f(\gamma(t)) - f(\gamma(0)) = \int_0^t \max_{d \in \mathcal{D}_f(\gamma(s))} \langle \dot{\gamma}(s), d \rangle \, \mathrm{d}s$ for any $t \geq 0$. It is worth mentioning that the most important choice of the conservative field $\mathcal{D}_f$ is the Clarke subdifferential of $f$. Moreover, as discussed in (Bolte & Pauwels, 2021; Castera et al., 2021; Davis et al., 2020), the class of path-differentiable functions are general enough to cover a wide range of objective functions in neural network training tasks, especially when the neural networks employ nonsmooth building blocks, such as the ReLU activation function. In addition, Bolte & Pauwels (2020; 2021) show that the outputs of AD algorithms in differentiating nonsmooth neural networks are contained in a conservative field of the loss function. Therefore, the concept of the conservative field is capable of characterizing the outputs of AD algorithms, which are implemented in training nonsmooth neural networks in practice.

Based on the stochastic approximation frameworks (Benaïm, 2006; Benaïm et al., 2005; Borkar, 2009; Davis et al., 2020), several existing works have investigated the convergence properties of stochastic subgradient methods in training nonsmooth neural networks. In particular, Bolte & Pauwels (2021); Davis et al. (2020) study the convergence properties of SGD and proximal SGD for minimizing nonsmooth path-differentiable functions. Moreover, (Castera et al., 2021) proposes the inertial Newton algorithm (INNA), which can be regarded as a variant of momentum-accelerated SGD method. Additionally, Le (2023); Ruszczyński (2020); Xiao et al. (2023b) establish the convergence properties of SGD with heavy-ball momentum, while Ding & Toh (2024) investigates the convergence of Bregman-type methods. Furthermore, Hu et al. (2022a;b) apply these methods to solve manifold optimization problems based on the constraint dissolving approach (Xiao et al., 2023c). In addition, Gürbüzbalaban et al. (2022); Ruszczynski (2021) design stochastic subgradient methods for solving multi-level nested optimization problems.

### 1.1.1 Challenges from non-diminishing stepsizes in Adam

With the concept of conservative field, Adam utilizes the following framework when applied to solve (UOP):

$$\begin{cases} g_k = d_k + \xi_{k+1}, \\ m_{k+1} = (1 - \theta_k)m_k + \theta_k(g_k + \sigma x_k), \\ v_{k+1} = (1 - \beta_k)v_k + \beta_k(g_k + \sigma x_k)^2, \\ x_{k+1} = x_k - \eta_k(\sqrt{v_{k+1}} + \varepsilon)^{-1} \odot m_{k+1}. \end{cases} \quad \text{(Adam)}$$

Here, $g_k$ is a stochastic subgradient of $f$ at $x_k$, in the sense that $d_k$ represents a possibly inexact evaluation of $\mathcal{D}_f(x_k)$, $\xi_{k+1}$ is a random vector characterizing the evaluation noise and $\sigma x_k$ is the weight decay term. The operators $\odot$ and $(\cdot)^p$ denote element-wise multiplication and element-wise $p$-th power of a given vector, respectively. The sequences $\{m_k\}$ and $\{v_k\}$, referred to as momentum terms and estimators respectively, are updated to track the first-order and second-order moments of $\{g_k + \sigma x_k\}$. The sequences $\{\eta_k\}$, $\{\theta_k\}$, and $\{\beta_k\}$ represent the stepsizes for the primal variables $\{x_k\}$, the parameters for the momentum terms $\{m_k\}$, and the parameters for the estimators $\{v_k\}$, respectively.

In the framework (Adam), the weight decay term is integrated with the function $f$ throughout the iterations. As a result, we can directly apply the existing convergence results on Adam to analyze the convergence properties of the framework (Adam). In particular, when $f$ is a nonsmooth path-differentiable function, (Xiao et al., 2023a) investigates the convergence of a class of Adam-family methods based on the frameworks proposed by (Benaïm et al., 2005; Bianchi et al., 2022; Davis et al., 2020). However, in the analysis of (Xiao et al., 2023a), the stepsizes and parameters sequences are assumed to be diminishing and single-timescale, in the sense that $\{\eta_k\}$, $\{\theta_k\}$ and $\{\beta_k\}$ converge to 0 at the same rate as $k$ goes to infinity.

Beyond the single-timescale scheme, some existing works (Reddi et al., 2018; Zhang et al., 2022; Jin et al., 2024) establish the convergence of Adam for continuously differentiable $f$ with $\{\theta_k\}$ and $\{\beta_k\}$ fixed as constants. In particular, (Zhang et al., 2022) proves that for any $\theta \in (0, 1)$ and $\eta_k = \mathcal{O}(1/\sqrt{k})$, there exists a sufficiently small $\beta$ that forces $\{x_k\}$ to stabilize within a neighborhood of the critical points of $g$. In addition, (Jin et al., 2024) establishes the asymptotic convergence convergence of Adam with relaxed conditions on the smoothness of $f$. However, their analyses are restricted to continuously differentiable objectives. Therefore, these results are not capable of explaining the convergence of Adam in a wide range of practical settings, where the neural networks are built from nonsmooth blocks. It is worth noting that (Xiao et al., 2023b) establishes the convergence of stochastic subgradient methods with two-timescale stepsizes. However, their analysis is restricted with SGD-type methods. To the best of our knowledge, this analysis cannot be extended to Adam-family methods.

Furthermore, in establishing convergence properties for stochastic subgradient methods, diminishing step sizes are commonly assumed as they facilitate the almost sure convergence of the iterates $\{x_k\}$ to critical points under various conditions (Benaïm et al., 2005; Bolte et al., 2022a; Bolte & Pauwels, 2021; Castera et al., 2021; Davis et al., 2020; Le, 2023; Ruszczyński, 2020; Xiao et al., 2023a;b). However, for Adam, the results in (Reddi et al., 2018; Zhang et al., 2022) indicate that even with diminishing step sizes, the iterates $\{x_k\}$ are only guaranteed to converge to a neighborhood of critical points. Moreover, Bianchi et al. (2022); Josz et al. (2023) show that for nonsmooth, path-differentiable objective functions and a fixed step size, SGD converges almost surely only to a neighborhood of the $\mathcal{D}_f$-stationary points of $f$. Their analysis, however, is limited to SGD and SGD with heavy-ball momentum and does not extend to Adam. Despite the widespread use of diminishing step size schemes in practice, it is both theoretically and practically important to investigate the convergence properties of Adam-family methods in the non-diminishing regime (i.e., when $\liminf_{k \to \infty} \eta_k > 0$).

### 1.1.2 Challenges from decoupling the weight decay term in Adam

Another challenge in solving (UOP) by Adam is related to the incorporation of the weight decay term. The conventional approach is to directly minimize $g$ by Adam, as is implemented in various computational frameworks. That is, the weight decay is coupled with the stochastic subgradients of $f$, in the sense that $f$ and the weight decay term $\frac{\sigma}{2} \|x\|^2$ are treated as an integrated function to be minimized. As demonstrated

in (Loshchilov & Hutter, 2019), Adam with coupled weight decay usually exhibits worse generalization performance than SGD. To address this issue, Loshchilov & Hutter (2019) suggests a novel method named AdamW, which decouples the weight decay term from the stochastic subgradients of $f$. The update schemes of AdamW can be summarized by the following framework:

$$\begin{cases} g_k = d_k + \xi_{k+1}, \\ m_{k+1} = (1 - \theta_k)m_k + \theta_k g_k, \\ v_{k+1} = (1 - \beta_k)v_k + \beta_k(g_k)^2, \\ x_{k+1} = x_k - \eta_k(\sqrt{v_{k+1}} + \varepsilon)^{-1} \odot m_{k+1} - \eta_k \sigma x_k. \end{cases} \tag{AdamW}$$

Here, Loshchilov & Hutter (2019) demonstrates that the weight decay is decoupled from the momentum terms $\{m_k\}$ and the estimators $\{v_k\}$, in the sense that the update schemes for $\{m_k\}$ and $\{v_k\}$ are independent of the weight decay parameter $\sigma$. Moreover, unlike (Adam), the weight decay term $\sigma x_k$ is not scaled by the preconditioner $(\sqrt{v_{k+1}} + \varepsilon)^{-1}$ in AdamW.

The AdamW, recognized for its superior generalization performance over Adam with coupled weight decay (i.e., the method in (Adam)), has become a popular choice in the training of neural networks (Loshchilov & Hutter, 2019), particularly in tasks such as image classification and language modeling. As shown in (Schaipp, 2023; Zhuang et al., 2022), AdamW can be interpreted as a proximal approximation of Adam, with regularization handled via proximal operators. However, compared with Adam, the convergence properties of AdamW and its proximal variants (Schaipp, 2023; Zhuang et al., 2022) remain relatively unexplored. As suggested in (Loshchilov & Hutter, 2019; Zhou et al., 2024), AdamW iterates by taking a descent step towards a dynamically adjusted surrogate function $f(x) + \frac{\sigma}{2} \langle x, (\sqrt{v_{k+1}} + \varepsilon) \odot x \rangle$ in the $k$-th iteration, thereby lacking a clearly defined objective function to minimize. As a result, only the paper by (Zhou et al., 2024) has established the convergence properties of AdamW for continuously differentiable $f$. In (Zhou et al., 2024), the stationarity of AdamW is measured by $\left\| \nabla f(x) + \sigma(\sqrt{v_{k+1}} + \varepsilon) \odot x \right\|$. As the estimators $\{v_k\}$ evolves over iterations and may not converge, the proposed stationarity measure is at best an approximation of the standard notion of stationarity. More importantly, the analysis in (Zhou et al., 2024) relies on the differentiability of $f$, and cannot be extended to analyze the convergence of AdamW for nonsmooth cases. Consequently, the results presented in (Zhou et al., 2024) do not sufficiently explain the convergence of AdamW in real-world training tasks, where the neural networks are typically nonsmooth.

Given that Adam-family methods with coupled weight decay usually perform less effectively than AdamW, and considering that AdamW lacks convergence guarantees in training nonsmooth neural networks, we are driven to raise the following question:

> Can we design Adam-family methods with decoupled weight decay that have convergence guarantees with non-diminishing stepsizes, in the context of training nonsmooth neural networks?

## 1.2 Contributions

The contributions of our paper are summarized as follows.

- **A novel framework with decoupled weight decay**

  In this paper, motivated by AdamW, we propose a novel framework for Adam-family methods with decoupled weight decay (AFMDW),

  $$\begin{cases} g_k = d_k + \xi_{k+1}, \\ m_{k+1} = (1 - \theta_k)m_k + \theta_k g_k, \\ \text{Choose the estimator } v_{k+1}, \\ x_{k+1} = x_k - \eta_k H(v_{k+1}) \odot (m_{k+1} + \sigma x_k). \end{cases} \tag{AFMDW}$$

  Here, $d_k$ is an approximated evaluation of $\mathcal{D}_f(x_k)$, while $\xi_{k+1}$ is the corresponding evaluation noise of $d_k$. Therefore, $g_k$ represents the stochastic subgradients of $f$ at $x_k$. Moreover, the sequences

$\{\eta_k\}$ and $\{\theta_k\}$ are stepsizes for the variables $\{x_k\}$ and parameters for the momentum terms $\{m_k\}$, respectively. Furthermore, $H : \mathbb{R}^n \to \mathbb{R}^n$ is the mapping that determines how we construct the preconditioner based on $v_{k+1}$. As the framework (AFMDW) is designed to minimize (UOP), both the momentum term $m_{k+1}$ and the weight decay term $\sigma x_k$ are scaled by $H(v_{k+1})$ in (AFMDW), distinguishing it from AdamW.

- **Convergence analysis**

  We establish the global convergence of the framework (AFMDW) under mild conditions with non-diminishing stepsizes. When the noises $\{\xi_k\}$ correspond to random reshuffling (RR), and the estimator $\{v_k\}$ is updated as in (AdamW) with non-diminishing $\{\eta_k\}$ and $\{\beta_k\}$, we prove that with sufficiently small but non-diminishing $\{\theta_k\}$, the sequence $\{x_k\}$ could stabilize within a neighborhood of the critical points of (UOP). In addition, when we further assume $\{\theta_k\} \to 0$, we prove that the sequence $\{x_k\}$ converges to the critical points of (UOP) almost surely. Moreover, by employing single-timescale scheme in (AFMDW), we prove that with sufficiently small $\{\eta_k\}$, the sequence $\{x_k\}$ stabilizes within a neighborhood of the critical points of (UOP).

  Furthermore, we extend the convergence analysis of the framework (AFMDW) with diminishing stepsizes and with replacement sampling (WRS), and establish the almost sure convergence to critical points of (UOP). Table 1 presents a brief comparison of our results with existing works on the convergence of stochastic subgradient methods.

Table 1: A brief comparison of our results and existing works on the convergence of stochastic subgradient methods.

| Result | Sampling method | Update scheme | Stepsizes | Convergence | Guaranteed stability |
|---|---|---|---|---|---|
| Theorem 3.10 & 3.22 | WRS | Adam | Diminishing | Almost sure | Y |
| Theorem 3.13 | RR | Adam | Constant | Almost sure | Y |
| (Josz et al., 2023) | RR | SGD | Constant | Almost sure | Y |
| (Bianchi et al., 2022) | WRS | SGD | Constant | High probability | Y |
| (Xiao et al., 2023a) | WRS | Adam | Diminishing | Almost sure | N |

- **Advantages in incorporating weight decay into Adam**

  We demonstrate that the framework (AFMDW) encompasses (see Table 2 for details) a wide range of Adam-family methods, including SGD, Adam, AMSGrad, AdaBelief, AdaBound, Yogi. Therefore, our analysis provides convergence guarantees for these Adam-family methods in training nonsmooth neural networks.

  Moreover, compared with the non-convergence analysis of Adam in (Reddi et al., 2018; Zhang et al., 2022), our analysis illustrates that the incorporation of a weight decay term grants more flexibility on the choices of the parameters $\{\theta_k\}$ and $\{\beta_k\}$ for the framework (AFMDW). These results illustrate the great theoretical advantages of a weight decay term in the framework (AFMDW).

- **Numerical experiments**

  Based on our proposed framework (AFMDW), we develop a novel method named Adam with Decoupled Weight Decay (AdamD) and establish its convergence guarantees in training nonsmooth neural networks. We conduct numerical experiments in both image classification and language modeling tasks to assess the performance of our proposed AdamD. The results show that in image classification tasks, AdamD outperforms Adam and performs comparably to AdamW in both generalization and efficiency. In language modeling tasks, it demonstrates similar effectiveness to Adam and outperforms AdamW, highlighting its versatility and effectiveness across different tasks. Additionally, our numerical experiments illustrate that the sequence $\{\|y_k - x_k\|\}$ tends to 0, where $y_k$ is an auxiliary variable that approximates the dynamics of SGD. This validates our theoretical analysis that the proposed AdamD asymptotically approximates the SGD method. These results further demonstrate the promising potential of our proposed framework (AFMDW).

### 1.3 Organization

The rest of this paper is organized as follows. In Section 2, we define the notations used throughout the paper and present some basic concepts related to nonsmooth analysis and stochastic approximation. Section 3 presents the convergence properties of our proposed framework (AFMDW) with non-diminishing stepsizes $\{\eta_k\}$. Moreover, we extend these convergence properties to the framework (AFMDW) with single-timescale stepsizes. As an application of our theoretical analysis, we propose a new Adam-family method named Adam with Decoupled Weight Decay (AdamD) and establish its convergence properties in Section 4. In Section 5, we present the results of our numerical experiments that investigate the performance of the proposed AdamD in training nonsmooth neural networks. Some further discussions on AdamD are also presented in Section 5. Finally, we conclude the paper in the last section.

## 2 Preliminaries

### 2.1 Notations

For any vectors $x$ and $y$ in $\mathbb{R}^n$ and $\delta \in \mathbb{R}$, we denote $x \odot y$, $x^\delta$, $x/y$, $|x|$, $x + \delta$, $\sqrt{x}$ as the vectors whose $i$-th entries are given by $x_i y_i$, $x_i^\delta$, $x_i/y_i$, $|x_i|$, $x_i + \delta$, and $\sqrt{x_i}$, respectively. We denote $\mathbb{R}_+^n := \{x \in \mathbb{R}^n : x_i \geq 0 \text{ for any } 1 \leq i \leq n\}$. Moreover, for any subsets $\mathcal{X}, \mathcal{Y} \subset \mathbb{R}^n$, we denote $\mathcal{X} \odot \mathcal{Y} := \{x \odot y : x \in \mathcal{X}, y \in \mathcal{Y}\}$, $|\mathcal{X}| := \{|x| : x \in \mathcal{X}\}$ and $\|\mathcal{X}\| = \sup\{\|w\| : w \in \mathcal{X}\}$. In addition, for any $z \in \mathbb{R}^n$, we denote $z + \mathcal{X} := \{z\} + \mathcal{X}$ and $z \odot \mathcal{X} := \{z\} \odot \mathcal{X}$.

Furthermore, for any positive sequence $\{\theta_k\}$, we define $\lambda_0 := 0$, $\lambda_i := \sum_{k=0}^{i-1} \theta_k$ for $i \geq 1$, and $\Lambda(t) := \sup\{k \geq 0 : t \geq \lambda_k\}$. More explicitly, $\Lambda(t) = p$ if $\lambda_p \leq t < \lambda_{p+1}$ for any $p \geq 0$. In particular, $\Lambda(\lambda_p) = p$.

### 2.2 Noise model

In this subsection, we present some essential concepts from probability theory, which are necessary for the proofs in this paper.

**Definition 2.1.** *Let $(\Omega, \mathcal{F}, \mathbb{P})$ be a probability space. We say that $\{\mathcal{F}_k\}_{k \in \mathbb{N}}$ is a filtration if $\{\mathcal{F}_k\}$ is a collection of $\sigma$-algebras that satisfies $\mathcal{F}_0 \subseteq \mathcal{F}_1 \subseteq \cdots \subseteq \mathcal{F}_\infty \subseteq \mathcal{F}$.*

**Definition 2.2.** *We say that a stochastic series $\{\xi_k\}$ is a martingale difference sequence if the following conditions hold,*

- *The sequence of random vectors $\{\xi_k\}$ is adapted to the filtration $\{\mathcal{F}_k\}$,*

- *For each $k \geq 0$, almost surely, it holds that $\mathbb{E}[\|\xi_k\|] < \infty$ and $\mathbb{E}[\xi_k|\mathcal{F}_{k-1}] = 0$.*

*Moreover, we say that a martingale difference sequence $\{\xi_k\}$ is uniformly bounded if there exists a constant $M_\xi > 0$ such that $\sup_{k \geq 0} \|\xi_k\| \leq M_\xi$.*

In the following, we present the results in (Benaïm, 2006, Proposition 4.4), which controls the weighted summation of any uniformly bounded martingale difference sequence, and plays a crucial role in establishing the convergence properties for our proposed framework (AFMDW).

**Proposition 2.3** (Proposition 4.4 in (Benaïm, 2006)). *Suppose $\{\theta_k\}$ is a diminishing positive sequence of real numbers that satisfy $\lim_{k \to \infty} \theta_k \log(k) = 0$. Then for any $T > 0$, and any uniformly bounded martingale difference sequence $\{\xi_k\}$, almost surely it holds that*

$$\lim_{s \to \infty} \sup_{s \leq i \leq \Lambda(\lambda_s + T)} \left\| \sum_{k=s}^i \theta_k \xi_{k+1} \right\| = 0. \tag{1}$$

### 2.3 Nonsmooth analysis

In this subsection, we introduce some basic concepts in nonsmooth optimization, especially those related to the concept of conservative field (Bolte & Pauwels, 2021). Interested readers could refer to (Bolte & Pauwels, 2021; Davis et al., 2020) for more details.

We begin our introduction on the concept of Clarke subdifferential (Clarke, 1990), which plays an essential role in characterizing stationarity and the development of algorithms for nonsmooth optimization problems.

**Definition 2.4** ((Clarke, 1990)). *For any given locally Lipschitz continuous function $f : \mathbb{R}^n \to \mathbb{R}$ and any $x \in \mathbb{R}^n$, the Clarke subdifferential $\partial f$ is defined as*

$$\partial f(x) := \mathrm{conv}\left(\{d \in \mathbb{R}^n : x_k \to x, \nabla f(x_k) \to d\}\right). \tag{2}$$

Next we present a brief introduction on the concept of conservative field, which can be applied to characterize how nonsmooth neural networks are differentiated by automatic differentiation (AD) algorithms.

**Definition 2.5.** *A set-valued mapping $\mathcal{D} : \mathbb{R}^n \rightrightarrows \mathbb{R}^s$ is a mapping from $\mathbb{R}^n$ to a collection of subsets of $\mathbb{R}^s$. $\mathcal{D}$ is said to have a closed graph, or is graph-closed if the graph of $\mathcal{D}$, defined by*

$$\mathrm{graph}(\mathcal{D}) := \{(w, z) \in \mathbb{R}^n \times \mathbb{R}^s : w \in \mathbb{R}^n, z \in \mathcal{D}(w)\},$$

*is a closed subset of $\mathbb{R}^n \times \mathbb{R}^s$.*

**Definition 2.6.** *A set-valued mapping $\mathcal{D} : \mathbb{R}^n \rightrightarrows \mathbb{R}^s$ is said to be locally bounded if, for any $x \in \mathbb{R}^n$, there is a neighborhood $V_x$ of $x$ such that $\cup_{y \in V_x} \mathcal{D}(y)$ is bounded.*

Next, we present the definition of conservative field and its corresponding potential function.

**Definition 2.7.** *An absolutely continuous curve is a continuous mapping $\gamma : \mathbb{R}_+ \to \mathbb{R}^n$ whose derivative $\gamma'$ exists almost everywhere in $\mathbb{R}_+$ and $\gamma(t) - \gamma(0)$ equals the Lebesgue integral of $\gamma'$ between $0$ and $t$ for all $t \in \mathbb{R}_+$, i.e.,*

$$\gamma(t) = \gamma(0) + \int_0^t \gamma'(u)\mathrm{d}u, \qquad \text{for all } t \in \mathbb{R}_+.$$

**Definition 2.8** (Definition 1 in (Bolte & Pauwels, 2021)). *Let $\mathcal{D}$ be a graph-closed set-valued mapping from $\mathbb{R}^n$ to subsets of $\mathbb{R}^n$. We call $\mathcal{D}$ a conservative field whenever it has nonempty compact values, and for any absolutely continuous curve $\gamma : [0, 1] \to \mathbb{R}^n$ satisfying $\gamma(0) = \gamma(1)$, it holds that*

$$\int_0^1 \max_{v \in \mathcal{D}(\gamma(t))} \langle \gamma'(t), v \rangle \, \mathrm{d}t = 0. \tag{3}$$

*Here the integral is understood in the Lebesgue sense.*

It is important to note that any conservative field is locally bounded (Bolte & Pauwels, 2021, Remark 3). We now introduce the definition of potential function corresponding to a conservative field.

**Definition 2.9** (Definition 2 in (Bolte & Pauwels, 2021)). *Let $\mathcal{D}$ be a conservative field in $\mathbb{R}^n$. Then with any given $x_0 \in \mathbb{R}^n$, we can define a function $f : \mathbb{R}^n \to \mathbb{R}$ through the path integral*

$$f(x) = f(x_0) + \int_0^1 \max_{d \in \mathcal{D}(\gamma(t))} \langle \gamma'(t), d \rangle \, \mathrm{d}t = f(x_0) + \int_0^1 \min_{d \in \mathcal{D}(\gamma(t))} \langle \gamma'(t), d \rangle \, \mathrm{d}t \tag{4}$$

*for any absolutely continuous curve $\gamma$ that satisfies $\gamma(0) = x_0$ and $\gamma(1) = x$. The function $f$ is called a potential function for $\mathcal{D}$. We also say that $\mathcal{D}$ admits $f$ as its potential function, or that $\mathcal{D}$ is a conservative field for $f$.*

The following two lemmas characterize the relationship between conservative field and Clarke subdifferential.

**Lemma 2.10** (Theorem 1 in (Bolte & Pauwels, 2021)). *Let $f : \mathbb{R}^n \to \mathbb{R}$ be a potential function that admits $\mathcal{D}_f$ as its conservative field. Then $\mathcal{D}_f(x) = \{\nabla f(x)\}$ almost everywhere.*

**Lemma 2.11** (Corollary 1 in (Bolte & Pauwels, 2021)). *Let $f : \mathbb{R}^n \to \mathbb{R}$ be a potential function that admits $\mathcal{D}_f$ as its conservative field. Then $\partial f$ is a conservative field for $f$, and for all $x \in \mathbb{R}^n$, it holds that*

$$\partial f(x) \subseteq \mathrm{conv}\left(\mathcal{D}_f(x)\right). \tag{5}$$

From the above two lemmas, we can conclude that the concept of conservative field can be regarded as a generalization of Clarke subdifferential. Therefore, conservative field can be applied to characterize stationarity, as illustrated in the following definition.

**Definition 2.12.** *Let $f : \mathbb{R}^n \to \mathbb{R}$ be a potential function that admits $\mathcal{D}_f$ as its conservative field. We say that $x$ is a $\mathcal{D}_f$-stationary point of $f$ if $0 \in \mathrm{conv}\,(\mathcal{D}_f(x))$. In particular, we say $x$ is a $\partial f$-stationary point of $f$ if $0 \in \partial f(x)$.*

As demonstrated in (Bolte & Pauwels, 2021), a conservative field can be regarded as a generalization of Clarke subdifferential. Therefore, a function is differentiable in the sense of conservative field if it admits a conservative field for which Definition 2.9 holds true. Such functions are called path-differentiable (Bolte & Pauwels, 2021, Definition 3), which is given below.

**Definition 2.13.** *Given a locally Lipschitz continuous function $f : \mathbb{R}^n \to \mathbb{R}$, we say that $f$ is path-differentiable if $f$ is the potential function of a conservative field on $\mathbb{R}^n$.*

It is worth mentioning that the class of path-differentiable functions is general enough to cover the objectives in a wide range of real-world problems. As shown in (Davis et al., 2020, Section 5.1), any Clarke regular function is path-differentiable. Beyond Clarke regular functions, another important class of path-differentiable functions are functions whose graphs are definable in an *o*-minimal structure (Davis et al., 2020, Definition 5.10). Usually, the *o*-minimal structure is fixed, and we simply call these functions definable. As demonstrated in (Van den Dries & Miller, 1996), any definable function admits a Whitney $C^s$ stratification (Davis et al., 2020, Definition 5.6) for any $s \geq 1$, hence is path-differentiable (Bolte & Pauwels, 2021; Davis et al., 2020). To characterize the class of definable functions, (Davis et al., 2020; Bolte & Pauwels, 2021; Bolte et al., 2022b) shows that numerous common activation functions and dissimilarity functions are all definable. Furthermore, since definability is preserved under finite summation and composition (Bolte & Pauwels, 2021; Davis et al., 2020), for any neural network built from definable blocks, its loss function is definable and thus belongs to the class of path-differentiable functions.

Moreover, (Bolte et al., 2007) shows that any Clarke subdifferential of definable functions is definable. Consequently, for any neural network constructed from definable blocks, the conservative field corresponding to the AD algorithms can be chosen as a definable set-valued mapping formulated by compositing the Clarke subdifferentials of all its building blocks (Bolte & Pauwels, 2021). The following proposition shows that the definability of $f$ and $\mathcal{D}_f$ leads to the nonsmooth Morse–Sard property (Bolte et al., 2007) for (UOP).

**Proposition 2.14** (Theorem 5 in (Bolte & Pauwels, 2021))**.** *Let $f$ be a potential function that admits $\mathcal{D}_f$ as its conservative field. Suppose both $f$ and $\mathcal{D}_f$ are definable over $\mathbb{R}^n$, then the set $\{f(x) : 0 \in \mathrm{conv}\,(\mathcal{D}_f(x))\}$ is finite.*

## 2.4 Differential inclusion and stochastic subgradient methods

In this subsection, we introduce some fundamental concepts related to the stochastic approximation technique that are essential for the proofs presented in this paper. The concepts discussed in this subsection are mainly from (Benaïm et al., 2005). Interested readers could refer to (Benaïm, 2006; Benaïm et al., 2005; Borkar, 2009; Davis et al., 2020) for more details on the stochastic approximation technique.

**Definition 2.15.** *For any locally bounded set-valued mapping $\mathcal{D} : \mathbb{R}^n \rightrightarrows \mathbb{R}^n$ that is nonempty compact convex valued and has closed graph, we say that an absolutely continuous path $x(t)$ in $\mathbb{R}^n$ is a solution for the differential inclusion*

$$\frac{\mathrm{d}x}{\mathrm{d}t} \in \mathcal{D}(x), \tag{6}$$

*with initial point $x_0$ if $x(0) = x_0$, and $\dot{x}(t) \in \mathcal{D}(x(t))$ holds for almost every $t \geq 0$.*

**Definition 2.16.** *For any given set-valued mapping $\mathcal{D} : \mathbb{R}^n \rightrightarrows \mathbb{R}^n$ and any constant $\delta \geq 0$, the set-valued mapping $\mathcal{D}^\delta$ is defined as*

$$\mathcal{D}^\delta(x) := \{w \in \mathbb{R}^n : \exists z \in \mathbb{B}_\delta(x), \mathrm{dist}(w, \mathcal{D}(z)) \leq \delta\}. \tag{7}$$

**Definition 2.17.** *Let $\mathcal{B} \subset \mathbb{R}^n$ be a closed set. A continuous function $\phi : \mathbb{R}^n \to \mathbb{R}$ is referred to as a Lyapunov function for the differential inclusion* (6) *with the stable set $\mathcal{B}$, if it satisfies the following conditions:*

1. *For any $\gamma$ that is a solution for* (6) *with $\gamma(0) \in \mathcal{B}$, it holds that $\phi(\gamma(t)) \leq \phi(\gamma(0))$ for any $t \geq 0$.*

2. *For any $\gamma$ that is a solution for* (6) *with $\gamma(0) \notin \mathcal{B}$, it holds that $\phi(\gamma(t)) < \phi(\gamma(0))$ for any $t > 0$.*

The following proposition illustrates that $f$ is a Lyapunov function for the differential inclusion $\frac{\mathrm{d}x}{\mathrm{d}t} \in -\mathcal{D}_f(x)$. The proof of the following proposition directly follows from (Bolte & Pauwels, 2021), hence is omitted for simplicity.

**Proposition 2.18.** *Suppose $f$ is a path-differentiable function $f$ that admits $\mathcal{D}_f$ as its conservative field. Then $f$ is a Lyapunov function for the differential inclusion $\frac{\mathrm{d}x}{\mathrm{d}t} \in -\mathcal{D}_f(x)$ with the stable set $\{x \in \mathbb{R}^n : 0 \in \mathcal{D}_f(x)\}$.*

**Definition 2.19.** *We say that an absolutely continuous function $\gamma$ is a perturbed solution to* (6) *if there exists a locally integrable function $u : \mathbb{R}_+ \to \mathbb{R}^n$, such that*

- *For any $T > 0$, it holds that $\lim_{t \to \infty} \sup_{0 \leq l \leq T} \left\| \int_t^{t+l} u(s) \, \mathrm{d}s \right\| = 0$.*

- *There exists $\delta : \mathbb{R}_+ \to \mathbb{R}$ such that $\lim_{t \to \infty} \delta(t) = 0$ and $\dot{\gamma}(t) - u(t) \in \mathcal{D}^{\delta(t)}(\gamma(t))$.*

Now consider the sequence $\{x_k\}$ generated by the following updating scheme,

$$x_{k+1} = x_k + \eta_k(d_k + \xi_k), \tag{8}$$

where $\{\eta_k\}$ is a diminishing positive sequence of real numbers. We define the (continuous-time) interpolated process of $\{x_k\}$ generated by (8) as follows.

**Definition 2.20.** *The (continuous-time) interpolated process of $\{x_k\}$ generated by* (8) *is the mapping $w : \mathbb{R}_+ \to \mathbb{R}^n$ such that*

$$w(\lambda_i + s) := x_i + \frac{s}{\eta_i}(x_{i+1} - x_i), \quad s \in [0, \eta_i). \tag{9}$$

*Here $\lambda_0 := 0$, and $\lambda_i := \sum_{k=0}^{i-1} \eta_k$ for $i \geq 1$.*

The following lemma is an extension of (Benaïm et al., 2005, Proposition 1.3), which allows for inexact evaluations of the set-valued mapping $\mathcal{D}$. It shows that the interpolated process of $\{x_k\}$ from (8) is a perturbed solution of the differential inclusion (6).

**Lemma 2.21.** *Let $\mathcal{D} : \mathbb{R}^n \rightrightarrows \mathbb{R}^n$ be a locally bounded set-valued mapping that is nonempty compact convex valued with closed graph. Suppose the following conditions hold in* (8)*:*

1. *For any $T > 0$, it holds that $\lim_{s \to \infty} \sup_{s \leq i \leq \Lambda(\lambda_s + T)} \left\| \sum_{k=s}^i \eta_k \xi_k \right\| = 0$.*

2. *There exist a positive sequence $\{\delta_k\}$ such that $\lim_{k \to \infty} \delta_k = 0$ and $d_k \in \mathcal{D}^{\delta_k}(x_k)$.*

3. *$\sup_{k \geq 0} \|x_k\| < \infty$, $\sup_{k \geq 0} \|d_k\| < \infty$.*

*Then the interpolated process of $\{x_k\}$ defined in* (9) *is a perturbed solution for* (6)*.*

The following theorem summarizes the results in (Benaïm et al., 2005), which illustrates the convergence of $\{x_k\}$ generated by (8). It is worth mentioning that Theorem 2.22 is directly derived from putting (Benaïm et al., 2005, Proposition 3.27) and (Benaïm et al., 2005, Theorem 3.6) together. Therefore, we omit the proof of Theorem 2.22 for simplicity.

**Theorem 2.22.** *Let $\mathcal{D} : \mathbb{R}^n \rightrightarrows \mathbb{R}^n$ be a locally bounded set-valued mapping that is nonempty compact convex valued with closed graph. For any sequence $\{x_k\}$, suppose there exist a continuous function $\phi : \mathbb{R}^n \to \mathbb{R}$ and a closed subset $\mathcal{B}$ of $\mathbb{R}^n$ such that*

1. *$\phi$ is bounded from below, and the set $\{\phi(x) : x \in \mathcal{B}\}$ has empty interior in $\mathbb{R}$.*

2. *$\phi$ is a Lyapunov function for the differential inclusion* (6) *that admits $\mathcal{B}$ as its stable set.*

3. *The interpolated process of $\{x_k\}$ is a perturbed solution of* (6).

*Then any cluster point of $\{x_k\}$ lies in $\mathcal{B}$, and the sequence $\{\phi(x_k)\}$ converges.*

Similar results under slightly different conditions can be found in (Borkar, 2009; Davis et al., 2020; Duchi & Ruan, 2018). Moreover, towards the convergence properties of (8) with potentially non-diminishing stepsizes, several recent works (Bianchi et al., 2022; Josz et al., 2023; Xiao et al., 2023b) provide convergence guarantees under more relaxed conditions. Interested readers could refer to those works for details.

## 3 Global Convergence

In this section, we prove the convergence properties of the framework (AFMDW) even though the sequence of stepsizes $\{\eta_k\}$ is assumed to be non-diminishing. The proofs are provided in the Appendix.

### 3.1 Basic assumptions

We first make the following assumptions on the quadratically regularized optimization problem (UOP).

**Assumption 3.1.** *1. $f$ is a path-differentiable function that admits a convex valued set-valued mapping $\mathcal{D}_f$ as its conservative field.*

2. *There exists a constant $L > 0$ and $\nu \in [0, 1)$, such that for any $x \in \mathbb{R}^n$, it holds that $\|\mathcal{D}_f(x)\| \leq L(1 + \|x\|^\nu)$.*

3. *The set $\{g(x) : 0 \in \mathcal{D}_f(x) + \sigma x\}$ has empty interior in $\mathbb{R}$.*

As discussed in Section 2.3, the class of path-differentiable functions covers a wide variety of objective functions in real-world applications. In particular, for a wide range of common neural networks, their loss functions are definable and thus path-differentiable, as demonstrated in (Bolte & Pauwels, 2021; Castera et al., 2021; Davis et al., 2020). As a result, Assumption 3.1(1) is mild in practice. Moreover, Assumption 3.1(2) imposes a growth condition on the conservative field. Furthermore, Assumption 3.1(3) is referred to as the nonsmooth weak Sard's property, which is commonly observed in various existing works (Bianchi & Rios-Zertuche, 2021; Bolte et al., 2022a; Bolte & Pauwels, 2021; Castera et al., 2021; Davis et al., 2020; Le, 2023) and is demonstrated to be mild in (Bolte & Pauwels, 2021; Castera et al., 2021; Davis et al., 2020).

Notice that the chain rule holds for conservative fields (Bolte & Pauwels, 2021, Lemma 5), and it is easy to verify that $g$ is a path-differentiable function that admits $\mathcal{D}_f(x) + \sigma x$ as its conservative field. Therefore, in the rest of the paper, we fix the conservative field $\mathcal{D}_g : \mathbb{R}^n \rightrightarrows \mathbb{R}^n$ for the objective function $g$ in (UOP) as:

$$\mathcal{D}_g(x) := \mathcal{D}_f(x) + \sigma x. \tag{10}$$

In the following lemma, we present some basic properties of $\mathcal{D}_g$. The proof of Lemma 3.2 follows straight-forwardly from (Bolte & Pauwels, 2021, Corollary 4), hence it is omitted for simplicity.

**Lemma 3.2.** *Suppose Assumption 3.1 holds. Then $g$ is a path-differentiable function, and $\mathcal{D}_g$ is a convex-valued graph-closed conservative field that admits $g$ as its potential function.*

We also need the following assumptions on the framework (AFMDW) to establish its convergence properties.

**Assumption 3.3.** *1. There exist constants $\varepsilon_v$ and $M_v$ with $0 < \varepsilon_v < M_v$, such that $\varepsilon_v \leq H(v_k) \leq M_v$ holds for any $k \geq 0$.*

2. *There exists a non-negative sequence $\{\delta_k\}$ such that $\lim_{k \to \infty} \delta_k = 0$ and $d_k \in \mathcal{D}_f^{\delta_k}(x_k)$.*

3. *The sequence of noises $\{\xi_k\}$ is a uniformly bounded martingale difference sequence. That is, there exists a constant $M_\xi$ such that almost surely, $\sup_{k \geq 0} \|\xi_k\| \leq M_\xi$, and $\mathbb{E}[\xi_{k+1}|\mathcal{F}_k] = 0$ for any $k \geq 0$.*

Here, we make some comments on the assumptions in Assumption 3.3. Assumption 3.3(1) assumes the uniform boundedness of $\{H(v_k)\}$, which is satisfied in various existing works as shown in Table 2. In addition, later in Section 3.2, we provide some sufficient conditions that guarantee the uniform boundedness of $\{x_k\}$. Assumption 3.3(2) characterizes how $d_k$ approximates $\mathcal{D}_f(x_k)$.

Furthermore, Assumption 3.3(3) assumes that the evaluation noises $\{\xi_k\}$ is a uniformly bounded martingale difference sequence. Consider the setting where $f(x) = \frac{1}{N}\sum_{i=1}^{N} f_i(x)$ for locally Lipschitz continuous and path-differentiable functions $\{f_i\}$ that admit $\{\mathcal{D}_{f_i}\}$ as their conservative fields (i.e., $f$ follows a finite-sum formulation), $\{x_k\}$ is uniformly bounded, and $g_k \in \mathcal{D}_{f_{i_k}}(x_k)$ with randomly chosen $i_k \in [N]$ by WRS. Then as demonstrated in (Bolte & Pauwels, 2021; Castera et al., 2021), there exists $\{\xi_k\}$ such that $g_k - \xi_{k+1} \in \mathcal{D}_f(x_k)$ and $\{\xi_k\}$ is a uniformly bounded martingale difference sequence. Therefore, we can conclude that Assumption 3.3(3) characteizes the evaluation noises when $f$ follows a finite-sum formulation and the indexes are sampled by WRS.

## 3.2 Uniform boundedness of $\{x_k\}$ and $\{v_k\}$

In this subsection, we present some sufficient and easy-to-verify conditions that guarantee the validity of uniform boundedness of $\{x_k\}$. The following proposition illustrates that under some mild global growth conditions for $f$ and the uniform boundedness of $\{H(v_k)\}$, the sequence $\{x_k\}$ is uniformly bounded.

**Proposition 3.4.** *Suppose Assumption 3.1 and Assumption 3.3 hold, and $\sup_{k \geq 0} \eta_k \leq \frac{1}{\sigma \varepsilon_v}$. Then for any initial point $(x_0, m_0, v_0)$, there exists a constant $Q > 0$ such that $\sup_{k \geq 0} \|x_k\| \leq Q$.*

Next, we discuss the uniform boundedness of the sequence $\{H(v_k)\}$. Apart from Assumption 3.1 and Assumption 3.3, we make the assumption on the global Lipschitz continuity of $f$, in the sense that

$$\sup_{x \in \mathbb{R}^n} \|\mathcal{D}_f(x)\| \leq M_f, \qquad \text{for some constant } M_f > 0. \tag{11}$$

Such an assumption is standard in various existing works. Table 2 lists some Adam-family methods, where the sequence $\{H(v_k)\}$ remains uniformly bounded under Assumption 3.1, Assumption 3.3, and equation (11).

Table 2: Different update schemes for $\{v_k\}$ in the framework (AFMDW) under Assumption 3.1, Assumption 3.3, and (11). Here $\varepsilon, c_l, c_u > 0$ are hyper-parameters for these Adam-family methods.

| Method | Update scheme for $\{v_k\}$ | Formulation for $H(v)$ | Choice of $(\varepsilon_v, M_v)$ |
|---|---|---|---|
| SGDW (Loshchilov & Hutter, 2019) | $v_{k+1} = (1-\beta_1)v_k + \beta_1 g_k^2$ | $1$ | $(1,1)$ |
| Adam (Kingma & Ba, 2015) | $v_{k+1} = (1-\beta_1)v_k + \beta_1 g_k^2$ | $(\sqrt{v}+\varepsilon)^{-1}$ | $(\frac{1}{(M_f+M_\xi)+\varepsilon}, \frac{1}{\varepsilon})$ |
| AMSGrad (Reddi et al., 2018) | $v_{k+1} = \max\{v_k, (1-\beta_1)v_k + \beta_1 g_k^2\}$ | $(\sqrt{v}+\varepsilon)^{-1}$ | $(\frac{1}{(M_f+M_\xi)+\varepsilon}, \frac{1}{\varepsilon})$ |
| Adamax (Kingma & Ba, 2015) | $v_{k+1} = \max\{\beta_1 v_k, |g_k| + \varepsilon\}$ | $(v)^{-1}$ | $(\frac{1}{(M_f+M_\xi)^2+\varepsilon}, \frac{1}{\varepsilon})$ |
| RAdam (Liu et al., 2019) | $v_{k+1} = (1-\beta_1)v_k + \beta_1 g_k^2$ | $(\sqrt{v}+\varepsilon)^{-1}$ | $(\frac{1}{(M_f+M_\xi)+\varepsilon}, \frac{1}{\varepsilon})$ |
| AdaBelief (Zhuang et al., 2020) | $v_{k+1} = (1-\beta_1)v_k + \beta_1(g_k - m_{k+1})^2$ | $(\sqrt{v}+\varepsilon)^{-1}$ | $(\frac{1}{2(M_f+M_\xi)+\varepsilon}, \frac{1}{\varepsilon})$ |
| AdaBound (Luo et al., 2019) | $v_{k+1} = (1-\beta_1)v_k + \beta_1 g_k^2$ | $\min\{c_l, \max\{c_u, v^{-\frac{1}{2}}\}\}$ | $(c_l, c_u)$ |
| Yogi (Zaheer et al., 2018) | $v_{k+1} = v_k - \beta_1\text{sign}(v_k - g_k^2) \odot g_k^2$ | $(\sqrt{v}+\varepsilon)^{-1}$ | $(\frac{1}{(M_f+M_\xi)+\varepsilon}, \frac{1}{\varepsilon})$ |

## 3.3 Convergence with non-diminishing stepsizes $\{\eta_k\}$

**Assumption 3.5.** *The sequences of stepsizes $\{\eta_k\}$ and momentum parameters $\{\theta_k\}$ satisfy*

$$\eta_{\max} := \sup_{k \geq 0} \eta_k < \min\left\{\frac{2}{\sigma M_v}, \frac{1}{\sigma \varepsilon_v}\right\}, \quad \eta_{\min} := \inf_{k \geq 0} \eta_k > 0, \quad and \quad \sum_{k=0}^{\infty} \theta_k = \infty. \tag{12}$$

We begin our theoretical analysis with Lemma 3.6, which shows that the sequence $\{m_k\}$ and $\{g_k\}$ are uniformly bounded. Lemma 3.6 directly follows from the uniform boundedness of $\{x_k\}$ in Proposition 3.4 and $\{\xi_k\}$ in Assumption 3.3(3) and the fact that $\mathcal{D}_f$ is locally bounded, hence we omit its proof for simplicity.

**Lemma 3.6.** *Suppose Assumption 3.1 and Assumption 3.3 hold. Then there exists a constant $M_d > 0$ such that $\sup_{k \geq 0} \{\|g_k\| + \|m_k\|\} \leq M_d$ holds almost surely.*

Lemma 3.7 illustrates that $\|\sigma x_k + m_k\| \to 0$ as the momentum parameter $\{\theta_k\}$ diminishes.

**Lemma 3.7.** *Suppose Assumption 3.1, Assumption 3.3, and Assumption 3.5 hold. Then for any $\{\theta_k\}$ satisfying $\lim_{k \to +\infty} \theta_k = 0$, we have that $\lim_{k \to +\infty} \|\sigma x_k + m_k\| = 0$ holds almost surely.*

From the proof of Lemma 3.7, it follows that the asymptotic behavior of $\|\sigma x_k + m_k\|$ can be controlled by $\{\theta_k\}$ as $k \to \infty$. Specifically, from equation (32), we have $\lim_{k \to \infty} \hat{\delta}_{k+1} = 0$ when $\lim_{k \to \infty} \theta_k = 0$. Consequently, for any $\varepsilon > 0$, there exists a threshold $\theta_{max} > 0$ such that, if $\limsup_{k \to \infty} \theta_k \leq \theta_{max}$, it follows that $\limsup_{k \to \infty} \|\sigma x_k + m_k\| \leq \varepsilon$. Moreover, the convergence of $\|\sigma x_k + m_k\|$ is faster as $\{\theta_k\}$ decreases more rapidly.

Based on the Lemma 3.7, let the auxiliary sequence $\{y_k\}$ be defined as

$$y_k := -\frac{1}{\sigma} m_k, \quad \text{for any } k \geq 0. \tag{13}$$

Then we can conclude that $\lim_{k \to \infty} \|y_k - x_k\| = 0$. More importantly, substituting (13) into the update scheme for $\{m_k\}$ in (AFMDW), we arrive at the following relation

$$y_{k+1} = y_k - \frac{\theta_k}{\sigma} \left( d_k + \sigma y_k + \xi_{k+1} \right). \tag{14}$$

In the following lemma, we prove that $d_k + \sigma y_k$ can be regarded as an approximated evaluation for $\mathcal{D}_g(y_k)$.

**Lemma 3.8.** *Suppose Assumption 3.1, Assumption 3.3, and Assumption 3.5 hold. Then let $\delta_k^\star := (1 + \sigma)\delta_k + \hat{\delta}_k$, it holds that*

$$d_k + \sigma y_k \in \mathcal{D}_g^{\delta_k^\star}(y_k), \tag{15}$$

*where $\hat{\delta}_k$ is defined in equation (32).*

We can conclude from Lemma 3.8 that the auxiliary sequence $\{y_k\}$ follows the differential inclusion,

$$y_{k+1} \in y_k - \frac{\theta_k}{\sigma} \left( \mathcal{D}_g^{\delta_k^\star}(y_k) + \xi_{k+1} \right). \tag{16}$$

This fact illustrates that the sequence $\{y_k\}$ can be viewed as a sequence generated by the SGD method for minimizing $g$. Therefore, in the following proposition, we prove that the interpolated process of the sequence $\{y_k\}$ is a perturbed solution of the following differential inclusion:

$$\frac{\mathrm{d}y}{\mathrm{d}t} \in -\mathcal{D}_g(y). \tag{17}$$

We first present the results for the case where the noise is induced by with-replacement sampling.

**Proposition 3.9.** *Suppose Assumption 3.1, Assumption 3.3 and Assumption 3.5 hold, and $\lim_{k \to +\infty} \theta_k \log(k) = 0$. Then the interpolated process of the sequence $\{y_k\}$ is a perturbed solution for the differential inclusion (17).*

In the following theorem, we prove the convergence properties of the framework (AFMDW).

**Theorem 3.10.** *Suppose Assumption 3.1, Assumption 3.3 and Assumption 3.5 hold, and $\lim_{k \to +\infty} \theta_k \log(k) = 0$. Then almost surely, any cluster point of the sequence $\{x_k\}$ is a $\mathcal{D}_g$-stationary point of $g$, and $\{g(x_k)\}$ converges.*

In the rest of this subsection, we aim to establish the **global stability** of the framework (AFMDW), where the noises $\{\xi_k\}$ correspond to random reshuffling. Therefore, we make the following assumptions on the momentum parameters $\{\theta_k\}$ and noises $\{\xi_k\}$.

**Assumption 3.11.** *There exists an integer $N > 0$ such that*

1. *For any nonnegative integers $i, j < N$, it holds that $\theta_{kN+i} = \theta_{kN+j}$ for any $k \in \mathbb{N}_+$.*

2. *For any $j \in \mathbb{N}_+$, almost surely, it holds that $\sum_{k=jN}^{(j+1)N-1} \xi_{k+1} = 0$.*

Here we make several remarks on Assumptions 3.11. Assumption 3.11(1) demonstrates that the momentum parameters $\{\theta_k\}$ remain constant within each epoch, which is a standard setting in neural network training tasks.

Moreover, Assumption 3.11(2) characterizes the random reshuffling sampling technique for computing the stochastic subgradients $\{g_k\}$ in (AFMDW). For example, consider the case where $f(x) = \frac{1}{N}\sum_{i=1}^{N} f_i(x)$, with each $f_i$ being path-differentiable and admitting a convex-valued conservative field $\mathcal{D}_{f_i}(x)$. At each iteration, we sample $i_k$ from $[N] := \{1, 2, \ldots, N\}$ such that $\{i_k : jN \le k < (j+1)N\} = [N]$ holds for any $j \ge 0$. We then compute $g_k \in \mathcal{D}_{f_{i_k}}(x_k)$. This setup corresponds to the standard random reshuffling framework.

For each $k \ge 0$, let $j_k \ge 0$ be such that $k \in [j_k N, (j_k + 1)N)$, it follows that $g_k \in \mathcal{D}_f(x_k) + \xi_{k+1}$, where $\xi_{k+1}$ represents the evaluation noise. Defining

$$\delta_k = 2N \sum_{j_k N \le l < (j_k+1)N} \|x_{l+1} - x_l\|, \tag{18}$$

we have $g_k \in \mathcal{D}_{f_{i_k}}(x_k) \subseteq \mathcal{D}_{f_{i_k}}^{\delta_k/(2N)}(x_{j_k N})$. Consequently, it holds that

$$\frac{1}{N} \sum_{j_k N \le l < (j_k+1)N} g_l \in \frac{1}{N} \sum_{j_k N \le l < (j_k+1)N} \mathcal{D}_{f_{i_l}}^{\delta_k/(2N)}(x_{j_k N}) \subseteq \mathcal{D}_f^{\delta_k/2}(x_{j_k N}) \subseteq \mathcal{D}_f^{\delta_k}(x_k).$$

Furthermore, it holds that $\frac{1}{N}\sum_{jN \le k < (j+1)N} \xi_{k+1} = 0$. Therefore, we conclude that Assumption 3.11(2) corresponds to settings where the stochastic subgradients $\{g_k\}$ are generated by the random reshuffling sampling technique. Furthermore, as shown later in Theorem 3.13, establishing the global stability of the iterates only requires $\{\delta_k\}$ to be sufficiently small rather than diminishing. This condition is ensured when $\{\eta_k\}$ is chosen sufficiently small, given the definition of $\{\delta_k\}$ in (18).

**Lemma 3.12.** *Suppose Assumption 3.3(3) and Assumption 3.11 hold for the sequence of noises $\{\xi_k\}$ and momentum parameters $\{\theta_k\}$. Then for any $\varepsilon > 0$ and $T > 0$, there exists $\theta_\varepsilon > 0$ such that for any $\{\theta_k\}$ satisfying $\limsup_{k \to +\infty} \theta_k \le \theta_\varepsilon$, almost surely, it holds that*

$$\limsup_{s \to +\infty} \sup_{s \le i \le \Lambda(\lambda(s)+T)} \left\| \sum_{k=s}^{i} \theta_k \xi_{k+1} \right\| \le \varepsilon. \tag{19}$$

Then we have the following theorem illustrating the global stability of the framework (AFMDW) with non-diminishing $\{\theta_k\}$.

**Theorem 3.13.** *Suppose Assumption 3.1, Assumption 3.3(1)(3), Assumption 3.5, and Assumption 3.11 hold. Then for any $\varepsilon > 0$, there exists $\hat{\theta} > 0$ and $\hat{\delta} > 0$, such that for any $\{\theta_k\}$ and $\{\delta_k\}$ satisfying $\limsup_{k \to +\infty} \delta_k \le \hat{\delta}$ and $\limsup_{k \to +\infty} \theta_k \le \hat{\theta}$, almost surely, it holds that*

$$\limsup_{k \to +\infty} \operatorname{dist}(x_k, \{x \in \mathbb{R}^n : 0 \in \mathcal{D}_g(x)\}) \le \varepsilon. \tag{20}$$

Theorem 3.13 implies that as long as the momentum parameters $\{\theta_k\}$ are sufficiently small, the sequence $\{x_k\}$ maintains stability, regardless of how the estimator $v_k$ is updated. When the estimator $v_k$ is updated in the manner of Adam, i.e., as a second-moment estimator, AdamD consistently works as long as the momentum parameter $\theta_k$ is kept small, irrespective of the choice of parameters for updating the second-moment estimator $v_k$. Therefore, our proposed AdamD offers greater flexibility in selecting momentum parameters and those associated with updating $v_k$. For a numerical illustration, refer to Figure 4 in Section 5.1.3.

Moreover, from the results in Theorem 3.13, we can prove that with diminishing $\{\theta_k\}$, the sequence $\{x_k\}$ can asymptotically find the stationary points of (UOP). The result is presented in the following corollary and is omitted for simplicity.

**Corollary 3.14.** *Suppose Assumption 3.1, Assumption 3.3, Assumption 3.5, and Assumption 3.11 hold, and $\lim_{k \to +\infty} \theta_k = 0$. Then almost surely, any cluster point of the sequence $\{x_k\}$ is a $\mathcal{D}_g$-stationary point of $g$, and $\{g(x_k)\}$ converges.*

### 3.4 Convergence with a single-timescale in $\{\eta_k\}$ and $\{\theta_k\}$

In this subsection, we investigate the convergence of the framework (AFMDW) when the sequences of stepsizes $\{\eta_k\}$ and momentum parameters $\{\theta_k\}$ are single-timescale in the sense that they diminish at the same rate.

The convergence properties presented in Section 3 suggest that the sequence $\{y_k\}$ asymptotically approximates the trajectory of the differential inclusion (17). One may conjecture that this phenomenon is attributable to the involvement of non-diminishing stepsizes $\{\eta_k\}$ in the framework (AFMDW).

However, in this section, we aim to show that when single-timescale stepsizes and momentum parameters are employed in the framework (AFMDW), the interpolated process of $\{y_k\}$ is still a perturbed solution of the differential inclusion (17). These theoretical results suggest that it is the decoupled weight decay that leads to the asymptotic approximation of the differential inclusion (17) in the framework (AFMDW), regardless of the timescale of the employed sequences $\{\eta_k\}$ and $\{\theta_k\}$.

The proof techniques in this section are motivated by the techniques in (Xiao et al., 2023a, Section 3). To prove the convergence of (AFMDW) with single-timescale sequences $\{\eta_k\}$ and $\{\theta_k\}$, we make the following assumptions.

**Assumption 3.15.**     *1. There exists a locally bounded mapping $W : \mathbb{R}^n \times \mathbb{R}^n \to \mathbb{R}^n_+$ and a prefixed constant $\tau_2 > 0$ such that the sequence of estimators $\{v_k\}$ follows the update scheme $v_{k+1} = v_k - \tau_2 \eta_k (v_k - W(g_k, m_{k+1}))$.*

   *2. The mapping $H : \mathbb{R}^n_+ \to \mathbb{R}^n_+$ is fixed as $H(v) = (\max\{v, 0\} + \varepsilon)^{-\frac{1}{2}}$ for a prefixed constant $\varepsilon > 0$.*

   *3. The sequences $\{\eta_k\}$ and $\{\theta_k\}$ are positive and satisfies*

$$\sum_{k=0}^{\infty} \eta_k = \infty, \quad \sum_{k=0}^{\infty} \theta_k = \infty, \quad \lim_{k \to \infty} \frac{\theta_k}{\eta_k} = \tau_1, \tag{21}$$

   *for a prefixed positive constant $\tau_1 \in \left[\frac{\tau_2}{4}, \infty\right)$.*

   *4. There exists a non-negative sequence $\{\delta_k\}$ such that $\lim_{k \to \infty} \delta_k = 0$ and $d_k \in \mathcal{D}_f^{\delta_k}(x_k)$.*

   *5. The sequence of noises $\{\xi_k\}$ is a uniformly bounded martingale difference sequence.*

Here we make some comments on Assumption 3.15. Assumption 3.15(4)(5) are identical to Assumption 3.3(2)(3), respectively. Assumption 3.15(1) characterizes how the estimators $\{v_k\}$ are updated. As discussed in (Barakat & Bianchi, 2021; Xiao et al., 2023a), Assumption 3.15(1) is general enough to include the update schemes for Adam, AdaBelief, AMSGrad, and Yogi. Moreover, Assumption 3.15(2) fixes the formulation of the mapping $H$, and Assumption 3.15(3) assumes that the stepsizes $\{\eta_k\}$ and momentum parameters $\{\theta_k\}$ in the framework (AFMDW) are single-timescale.

We begin our analysis with the following lemma, which shows the uniform boundedness of $\{m_k\}$ and $\{g_k\}$ directly from the uniform boundedness of $\{x_k\}$ in Proposition 3.4. As a result, we omit its proof for simplicity.

**Lemma 3.16.** *Suppose Assumption 3.1 and Assumption 3.15 hold. Then there exists a constant $M_d > 0$ such that $\sup_{k \geq 0} \|g_k\| + \|m_k\| \leq M_d$ holds almost surely.*

Next we present the following auxiliary lemma, which follows directly from the uniform boundedness of $\{x_k\}$, $\{m_k\}$ and $\{g_k\}$ in Lemma 3.16, together with the local boundedness of the mappings $\mathcal{D}_f$ and $W$.

**Lemma 3.17.** *Suppose Assumption 3.1 and Assumption 3.15 hold. Then there exists a constant $M_W > 0$ such that $\sup_{k \geq 0} \|W(g_k, m_{k+1})\| \leq M_W$ holds almost surely.*

Let $\mathcal{P}_+(v) := \max\{v, 0\}$, and $\mathcal{U}(x, m) := \{d \in \mathbb{R}^n_+ : \|d\| \leq M_W\}$. Consider the set-valued mapping $\mathcal{G} : \mathbb{R}^n \times \mathbb{R}^n \times \mathbb{R}^n \rightrightarrows \mathbb{R}^n \times \mathbb{R}^n \times \mathbb{R}^n$ defined by

$$\mathcal{G}(x, m, v) := \begin{bmatrix} (\mathcal{P}_+(v) + \varepsilon)^{-\frac{1}{2}} \odot (m + \sigma x) \\ \tau_1 m - \tau_1 \mathcal{D}_f(x) \\ \tau_2 v - \tau_2 \mathcal{U}(x, m) \end{bmatrix}, \tag{22}$$

and the following differential inclusion:

$$\left( \frac{\mathrm{d}x}{\mathrm{d}t}, \frac{\mathrm{d}m}{\mathrm{d}t}, \frac{\mathrm{d}v}{\mathrm{d}t} \right) \in -\mathcal{G}(x, m, v). \tag{23}$$

In the following lemma, we prove that the set-valued mapping $\mathcal{G}$ is capable of characterizing the update direction of $\{(x_k, m_k, v_k)\}$ in the framework (AFMDW). The proof straightforwardly follows from Lemma 3.17, hence we omit it for simplicity.

**Lemma 3.18.** *Suppose Assumption 3.1 and Assumption 3.15 hold. Then the inclusion*

$$v_{k+1} \in v_k - \tau_2 \eta_k (v_k - \mathcal{U}(x_k, m_k)) \tag{24}$$

*holds for any $k \geq 0$. Furthermore, $\sup_{k \geq 0} \|v_{k+1}\| < \infty$ holds almost surely.*

Let $\partial \mathcal{P}_+$ be the generalized Jacobian of the mapping $\mathcal{P}_+$, and define the function $h : \mathbb{R}^n \times \mathbb{R}^n \times \mathbb{R}^n \to \mathbb{R}$ as

$$h(x, m, v) = f(x) + \frac{\sigma}{2} \|x\|^2 + \frac{1}{2\tau_1} \left\langle m + \sigma x, (\mathcal{P}_+(v) + \varepsilon)^{-\frac{1}{2}} \odot (m + \sigma x) \right\rangle. \tag{25}$$

The next Lemma 3.19 presents the formulation of the conservative field of $h$.

**Lemma 3.19.** *Suppose Assumption 3.1 and Assumption 3.15 hold. Then $h$ is a potential function that admits*

$$\mathcal{D}_h(x, m, v) = \begin{bmatrix} \mathcal{D}_f(x) + \sigma x + \frac{\sigma}{\tau_1}(\mathcal{P}_+(v) + \varepsilon)^{-\frac{1}{2}} \odot (m + \sigma x) \\ \frac{1}{\tau_1}(\mathcal{P}_+(v) + \varepsilon)^{-\frac{1}{2}} \odot (m + \sigma x) \\ -\frac{1}{4\tau_1}(m + \sigma x)^2 \odot (\mathcal{P}_+(v) + \varepsilon)^{-\frac{3}{2}} \odot \partial \mathcal{P}_+(v) \end{bmatrix} \tag{26}$$

*as its conservative field.*

**Proposition 3.20.** *Suppose Assumption 3.1 and Assumption 3.15 hold. Then $h$ is a Lyapunov function for the differential inclusion (23) with the stable set $\{(x, m, v) \in \mathbb{R}^n \times \mathbb{R}^n \times \mathbb{R}^n : 0 \in \mathcal{D}_g(x), m + \sigma x = 0\}$.*

In the next proposition, we show that the interpolated process of the sequence $\{(x_k, m_k, v_k)\}$ is a perturbed solution to the differential inclusion (23).

**Proposition 3.21.** *Suppose Assumption 3.1 and Assumption 3.15 hold, and $\lim_{k \to +\infty} \eta_k \log(k) = 0$. Then almost surely, the interpolated process of $\{(x_k, m_k, v_k)\}$ is a perturbed solution for the differential inclusion (23).*

In the following theorem, we present the convergence properties of the sequence $\{(x_k, m_k, v_k)\}$, and prove that $\lim_{k \to \infty} \|m_k + \sigma x_k\| = 0$ almost surely.

**Theorem 3.22.** *Suppose Assumption 3.1 and Assumption 3.15 hold, and $\lim_{k \to +\infty} \eta_k \log(k) = 0$. Then for the sequence $\{(x_k, m_k, v_k)\}$ generated by the framework (AFMDW), almost surely, it holds that*

1. *any cluster point of the sequence $\{x_k\}$ is a $\mathcal{D}_g$-stationary point of $g$;*

2. *$\lim_{k \to \infty} \|m_k + \sigma x_k\| = 0$;*

3. *the sequence of function values $\{g(x_k)\}$ converges.*

Theorem 3.22 illustrates that $\lim_{k\to\infty} \|x_k - y_k\| = 0$. Therefore, substituting the formulation of $\{y_k\}$ in (13) into the update scheme of $\{m_k\}$ in the framework (AFMDW), we conclude that $\{y_k\}$ follows the same scheme as (14). Together with the fact that $\lim_{k\to\infty} \|x_k - y_k\| = 0$, based on the same proof techniques as in Lemma 3.8, we can conclude that there exists a sequence of non-negative random variables $\{\tau_k\}$ such that $\lim_{k\to\infty} \tau_k = 0$ holds almost surely, and

$$y_{k+1} \in y_k - \frac{\theta_k}{\sigma}(\mathcal{D}_g^{\tau_k}(y_k) + \xi_{k+1}).$$

Then we have the following corollary showing that the interpolated process of the sequence $\{y_k\}$ is a perturbed solution of the differential inclusion (17). The proof of Corollary 3.23 is the same as Proposition 3.9, hence is omitted for simplicity.

**Corollary 3.23.** *Suppose Assumption 3.1 and Assumption 3.15 hold. Then the interpolated path of the sequence $\{y_k\}$ is a perturbed solution of the differential inclusion* (17).

## 4 Application: Adam with Decoupled Weight Decay

In this section, we propose a novel variant of Adam, which is named as Adam with decoupled weight decay (AdamD). As an application of our theoretical analysis in Section 3, we show the convergence properties of AdamD directly from the results in Theorem 3.10 and Theorem 3.22.

Throughout this section, we focus on the settings where $f$ in (UOP) takes the following finite-sum formulation:

$$f(x) = \frac{1}{N}\sum_{i=1}^{N} f_i(x). \tag{27}$$

Here we make the following assumptions on the functions $\{f_i : i \in [N]\}$ in (27).

**Assumption 4.1.** *1. For each $i \in [N]$, $f_i$ is a definable function that admits a definable set-valued mapping $\mathcal{D}_{f_i}$ as its conservative field.*

*2. $\sup_{i\in[N],\ x\in\mathbb{R}^n} \|\mathcal{D}_{f_i}(x)\| < \infty$.*

*3. $f$ is bounded from below.*

As demonstrated in (Bolte & Pauwels, 2021), for any neural network that is built from definable blocks, the conservative field corresponds the AD algorithms is a definable set-valued mapping. Hence, we can conclude that Assumption 4.1(1) can be satisfied in a wide range of training tasks. Assumption 4.1(2) assumes the Lipschitz continuity of the function $f$, which is common in various existing works (Barakat & Bianchi, 2021; Guo et al., 2021; Shi et al., 2021; Zhang et al., 2022).

Moreover, (Bolte et al., 2021, Corollary 4) illustrates that $f$ is a path-differentiable function and admits $\frac{1}{N}\sum_{i=1}^{N}\mathcal{D}_{f_i}$ as its conservative field. Therefore, in the rest of this section, we choose the conservative field $\mathcal{D}_f$ as

$$\mathcal{D}_f(x) = \mathrm{conv}\left(\frac{1}{N}\sum_{i=1}^{N}\mathcal{D}_{f_i}(x)\right). \tag{28}$$

The detailed AdamD method is presented in Algorithm 1. In our proposed AdamD method, the weight decay term $\sigma x_k$ is decoupled from the update schemes for $\{m_k\}$ and $\{v_k\}$. In particular, the estimators $\{v_k\}$ are updated as an exponential moving average over $\{g_k^2\}$ with parameter $\beta \in (0,1)$. We note that AdamP introduced in Schaipp (2023) employs a proximal operator to handle the $L_2$ regularizer term, derived from a first-order Taylor expansion with a variable metric. In practice, its updates yield approximately the same coefficients for $x_k$ and $m_k$, as our AdamD method. However, to the best of our knowledge, no convergence result is available for AdamP, even in the smooth case.

Based on the convergence properties of the framework (AFMDW) presented in Theorem 3.10, the following theorem establishes the convergence properties of Algorithm 1 with non-diminishing $\{\eta_k\}$.

**Algorithm 1** Adam with decoupled weight decay (AdamD) for nonsmooth problem (UOP).

**Require:** Initial point $x_0 \in \mathbb{R}^n$, $m_0 \in \mathbb{R}^n$, $v_0 \in \mathbb{R}^n_+$, weight decay parameter $\sigma > 0$, safeguard parameter $\varepsilon > 0$, stepsize $\eta \le \frac{\varepsilon}{\sigma}$, and $\beta \in (0, 1)$;

1: Set $k = 0$;
2: **while** not terminated **do**
3:     Independently sample $i_k$ from $[N]$, and compute $g_k \in \mathcal{D}_{f_{i_k}}(x_k)$;
4:     Update momentum:
$$m_{k+1} = (1 - \theta_k)m_k + \theta_k \, g_k;$$
5:     Update estimator:
$$v_{k+1} = (1 - \beta)v_k + \beta \, g_k^2;$$
6:     Update:
$$x_{k+1} = x_k - \eta \,\, (\sqrt{v_{k+1}} + \varepsilon)^{-1} \odot (m_{k+1} + \sigma x_k);$$
7:     Set $k \leftarrow k + 1$;
8: **end while**
9: **return** $x_k$.

**Algorithm 2** AdamW (Loshchilov & Hutter, 2019).

**Require:** Initial point $x_0 \in \mathbb{R}^n$, $m_0 \in \mathbb{R}^n$, $v_0 \in \mathbb{R}^n_+$, weight decay parameter $\sigma > 0$, safeguard parameter $\varepsilon > 0$, $\theta \in (0, 1)$, and $\beta \in (0, 1)$;

1: Set $k = 0$;
2: **while** not terminated **do**
3:     Independently sample $i_k$ from $[N]$, and compute $g_k \in \mathcal{D}_{f_{i_k}}(x_k)$;
4:     Update momentum:
$$m_{k+1} = (1 - \theta)m_k + \theta \, g_k;$$
5:     Update estimator:
$$v_{k+1} = (1 - \beta)v_k + \beta \, g_k^2;$$
6:     Update:
$$x_{k+1} = x_k - \eta \left( (\sqrt{v_{k+1}} + \varepsilon)^{-1} \odot m_{k+1} + \sigma x_k \right);$$
7:     Set $k \leftarrow k + 1$;
8: **end while**
9: **return** $x_k$.

**Theorem 4.2.** *Suppose Assumption 3.5 and Assumption 4.1 hold. Moreover, we assume that the momentum parameters $\{\theta_k\}$ is a positive sequence that satisfies $\lim_{k \to \infty} \theta_k \log(k) = 0$. Then almost surely, any cluster point of $\{x_k\}$ in Algorithm 1 is a $\mathcal{D}_g$-stationary point of $g$, and the sequence $\{g(x_k)\}$ converges.*

In the following theorem, we establish the convergence properties for Algorithm 1 when it is equipped with single-timescale stepsizes. The results in Theorem 4.3 are direct consequences of Theorem 3.22. Hence, we omit its proof for simplicity.

**Theorem 4.3.** *Suppose Assumption 4.1 holds. Moreover, we assume that*

1. *The stepsizes $\eta$ and $\beta$ are replaced by $\eta_k$ and $\beta_k$ respectively in Algorithm 1;*

2. *There exists constants $\tau_2 \ge 4\tau_1 > 0$ such that $\theta_k = \tau_1 \eta_k$ and $\beta_k = \tau_2 \eta_k$ hold for any $k \ge 0$. Moreover, the sequence $\{\eta_k\}$ satisfies $\sum_{k=0}^{\infty} \eta_k = \infty$ and $\lim_{k \to \infty} \eta_k \log(k) = 0$.*

3. *In Step 6 of Algorithm 1, the sequence $\{x_k\}$ follows the update scheme*
$$x_{k+1} = x_k - \eta_k (v_{k+1} + \varepsilon)^{-\frac{1}{2}} \odot (m_{k+1} + \sigma x_k).$$

*Then almost surely, any cluster point of $\{x_k\}$ in Algorithm 1 is a $\mathcal{D}_g$-stationary point of $g$, and the sequence $\{g(x_k)\}$ converges.*

## 5 Numerical Experiments

In this section, we conduct numerical experiments to demonstrate the effectiveness of AdamD in the context of image classification and language modeling tasks. We compare AdamD with the most popular adaptive algorithms used for training neural networks, i.e. Adam and AdamW. All experiments are conducted using an NVIDIA RTX 3090 Ti GPU and are implemented in Python 3.9 with PyTorch 1.12.0.

### 5.1 Implementations of AdamD

In our numerical experiments, we focus on two key tasks: image classification employing Convolutional Neural Networks (CNNs) and language modeling using Long Short-Term Memory (LSTM) networks (Hochreiter & Schmidhuber, 1997). Specifically, our image classification experiments include the deployment of well-established architectures, namely Resnet34 (He et al., 2016) and Densenet121 (Huang et al., 2018), to train the CIFAR-10 and CIFAR-100 datasets (Krizhevsky et al., 2009). Our language modeling experiments focus on LSTM networks applied to the Penn Treebank dataset (Marcus et al., 1993). It is worth noting that AdamW typically demonstrates superior generalization performance when used to train CNNs for image classification tasks. For training LSTMs, prior studies such as (Ding et al., 2023; Loshchilov & Hutter, 2019; Zhuang et al., 2020) have observed that Adam exhibits better generalization capacity than AdamW.

#### 5.1.1 CNNs on image classification

In all our experiments on image classification, we train the models consistently for 200 epochs, employing a batch size of 128. At the 150th epoch, we reduce the step size by a factor of 0.1. This step size reduction schedule is a prevalent practice in contemporary deep neural network training. It is helpful to accelerate the convergence of the optimization algorithm, and to enhance generalization capacity. Similar strategies can be observed in previous works, such as (He et al., 2016; Zhuang et al., 2020). We adopt the following hyperparameter settings for the tested algorithms. For the weight decay parameter, we consider values in $\sigma \in \{5 \times 10^{-3}, 10^{-3}, 5 \times 10^{-4}, 10^{-4}\}$. By fixing $\sigma$ first, we ensure that all methods solve the same minimization problem. With $\sigma$ fixed, we then perform a grid search over the learning rate $\eta$ for AdamD, Adam, and AdamW using $\eta \in \{5 \times 10^{-5}, 10^{-4}, 5 \times 10^{-4}, 10^{-3}, 5 \times 10^{-3}, 10^{-2}, 5 \times 10^{-2}, 10^{-1}\}$. Other parameters are set as follows:

- Adam/AdamW: We set $\varepsilon = 10^{-8}$, $\theta_k = 10^{-1}$ and $\beta = 10^{-3}$ as the default setting in Pytorch.

- AdamD: We set $\theta_s = \frac{\theta_0}{(\log(s+2))^{\frac{3}{2}}}$, with $s$ representing the epoch number. Within the $s$-th epoch, $\theta_k$ takes the constant value $\theta_s$. Under this setting, we can easily verify that $\theta_k = o(\frac{1}{\log k})$. Here, we set the initial momentum parameter to $\theta_0 = 10^{-1}$, the second moment parameter to $\beta = 10^{-3}$ and the regularization parameter to $\varepsilon = 10^{-8}$, which are the same as the default settings in PyTorch for Adam/AdamW.

In Step 6 of Algorithm 1, the coefficient associated with $x_k$ is given by $1 - \eta\sigma \left(\sqrt{v_{k+1}} + \varepsilon\right)^{-1}$. Note that as training progresses, $\sqrt{v_{k+1}} + \varepsilon$ tends to decrease, which can make the coefficient excessively small when the weight decay $\sigma$ is relatively large. In our experiments, larger weight decay parameters (exceeding $10^{-5}$) enhance generalization in image classification, whereas smaller weight decay parameters (not exceeding $10^{-5}$) yield better performance for LSTM training, as discussed in a later subsection. This trend is also observed in (Luo et al., 2019; Zhuang et al., 2020; Li et al., 2022). To enhance AdamD's performance in image classification, where larger weight decay typically yields better generalization, we adopt a clipping strategy for the preconditioner similar to that used in Adabound (Luo et al., 2019). Specifically, we define $\hat{\sigma}_k := \text{Clip}\left(\sigma(\sqrt{v_{k+1}} + \varepsilon)^{-1}, c_l, c_u\right)$, thereby balancing the coefficient, now expressed as $1 - \eta\hat{\sigma}_k$. Here, the clipping function is defined as $\text{Clip}(z, c_l, c_u) := \max\{\min\{z, c_u\}, c_l\}$, and we choose $(c_l, c_u) = (10^{-2}, 10^2)$.

Figure 1 presents numerical results obtained with a fixed weight decay parameter $\sigma = 5 \times 10^{-3}$ and the best-tuned learning rate $\eta$ that achieves the highest test accuracy, ensuring that all methods minimize the same objective function. Additional experimental results and further details are provided in Appendix B. As shown in the first row, all methods can achieve 100% training accuracy across four tasks. In the second row, AdamD consistently attains the highest test accuracy compared with Adam and AdamW, while Adam's generalization performance is inferior. This may be due to AdamD's asymptotic approximation to SGD with weight decay, which has been shown to perform better in image classification (Luo et al., 2019; Zhuang et al., 2020). These findings underscore the importance of weight decoupling when solving the quadratically regularized problem defined in (UOP).

To verify the results in Lemma 3.7, we also present a plot of $\|x_k + \sigma m_k\|$ as shown in Figure 2. When $\theta_k$ adheres to a decay schedule described by $\mathcal{O}(k^{-\gamma})$, (32) and basic calculus imply that $\|\sigma x_k + m_k\|$ exhibits

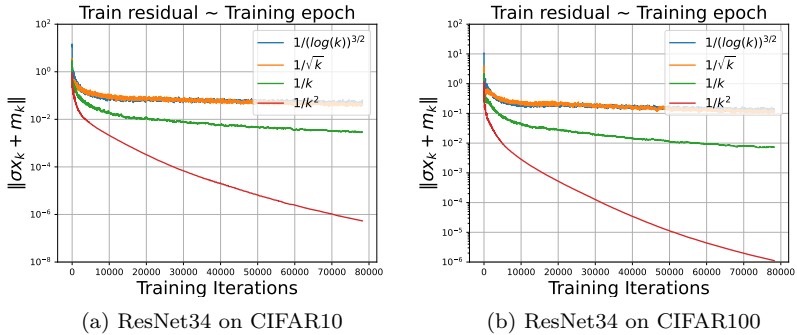

Figure 1: ResNet34 and DenseNet121 on CIFAR10 and CIFAR100 datasets. Stepsize is reduced to 0.1 times of the original value at the 150th epoch.

(a) ResNet34 on CIFAR10  (b) DenseNet on CIFAR10  (c) ResNet34 on CIFAR100  (d) DenseNet on CIFAR100

(a) ResNet34 on CIFAR10  (b) ResNet34 on CIFAR100

Figure 2: $\|m_k + \sigma x_k\|$ under different decay rates of $\{\theta_k\}$. The stepsizes for updating $\{x_k\}$ are fixed.

an asymptotic behavior of $\mathcal{O}\left(k^{-\gamma}\right)$. The results in Figure 2 are consistent with our theoretical analysis that $\{\|m_k + \sigma x_k\|\}$ converges to 0, or equivalently $\{\|x_k - y_k\|\}$ converges to 0. Notably, larger values of $\gamma$ correspond to a more rapid decline in $\|\sigma x_k + m_k\|$.

### 5.1.2   LSTMs on language modeling

In all our language modeling experiments, we train our models for 200 epochs using a batch size of 128. We employ a step size reduction strategy that decreases the learning rate to 0.1 times its previous value twice during training, specifically at the 75th and 150th epochs. These settings follow the commonly used experimental setup for training LSTMs, as demonstrated in previous works (Chen et al., 2021; Zhuang et al., 2020). This strategy accelerates convergence and enhances the generalization capacity of the optimization algorithm. We adopt the following hyperparameter settings for the tested algorithms. For the weight decay parameter, we consider values in $\sigma \in \{10^{-4}, 10^{-5}, 10^{-6}\}$. By first fixing $\sigma$, we ensure that all methods minimize the same objective function. With $\sigma$ fixed, we perform a grid search over the learning rate $\eta$ for AdamD, Adam, and AdamW using values in $\eta \in \{10^{-1}, 5 \times 10^{-2}, 10^{-2}, 5 \times 10^{-3}, 10^{-3}, 5 \times 10^{-4}\}$. In the training of LSTMs, we do not employ the clipping strategy since smaller weight decay parameters already yield good generalization performance. All other hyperparameter settings are identical to those used in Section 5.1.1. Figure 3 displays the numerical results obtained with a fixed $\sigma = 10^{-5}$, ensuring that all methods minimize the same objective function.

As shown in Figure 3, both AdamD and Adam outperform AdamW in terms of generalization. For 1-layer LSTMs, AdamD and Adam achieve comparable performance, whereas for deeper 2-layer and 3-layer LSTM models, AdamD reduces the test perplexity by at least 0.6 units compared to Adam. We summarize the tuned stepsizes in the experiments in Figure 1 and 3 in Table 3. Further experimental details can be found in Appendix B.

| | $\sigma$ : 5e-03 | | | | $\sigma$ : 1e-05 | | |
|---|---|---|---|---|---|---|---|
| | Task 1 | Task 2 | Task 3 | Task 4 | Task 5 | Task 6 | Task 7 |
| AdamD | 5e-03 | 5e-03 | 5e-03 | 5e-03 | 5e-03 | 5e-03 | 5e-03 |
| Adam | 1e-04 | 1e-04 | 1e-04 | 1e-04 | 5e-03 | 5e-03 | 5e-03 |
| AdamW | 1e-02 | 5e-04 | 5e-04 | 5e-04 | 1e-03 | 1e-03 | 1e-03 |

Table 3: Tuned stepsizes for varing tasks. Task 1: ResNet34 on CIFAR10; Task 2: DenseNet121 on CIFAR10; Task 3: ResNet34 on CIFAR100; Task 4: DenseNet121 on CIFAR100; Task 5: 1-layer LSTM; Task 6: 2-layer LSTM; Task 7: 3-layer LSTM.

### 5.1.3   Performance with different choices of $\{\theta_k\}$ and $\{\beta_k\}$

Finally, we investigate the performance of the AdamD method with different choices of its step sizes $\{\theta_k\}$ and $\{\beta_k\}$ for the momentum terms $\{m_k\}$ and $\{v_k\}$, respectively. In our numerical experiments, both sequences are fixed as constants, i.e., $\theta_k = \theta \in [0, 1]$ and $\beta_k = \beta \in [0, 1]$. The weight decay parameter is set to $5 \times 10^{-4}$, and the step size $\eta$ is tuned among $\{10^{-2}, 10^{-3}, 10^{-4}\}$ to maximize the proportion of training loss below 0.6. The regularization parameter for the second moment is $\varepsilon = 10^{-8}$, following the default settings in PyTorch for Adam.

Figure 4 shows that the light-colored regions (indicating lower training loss) for AdamW and AdamD are larger than that for Adam. As demonstrated in (Zhang et al., 2022), Adam can fail when $\theta$ is small and $\beta$ is large, as indicated by the red area in the bottom right portion of Figure 4(a). In contrast, the corresponding region for AdamD in Figure 4(c) remains light. These observations suggest that incorporating weight decay in AdamD enhances its robustness to the choices of $\theta$ and $\beta$ compared to Adam with coupled weight decay. Furthermore, the results support the theoretical findings in Theorem 3.13, which imply that if the momentum parameters $\theta_k$ are kept sufficiently small, the sequence $x_k$ remains stable. This stability is maintained with relaxed requirements on the updates of the estimator $v_k$, provided that the preconditioner $H(v_k)$ is bounded both below and above. An additional observation is that, while AdamD and AdamW exhibit similar overall robustness, the favorable regions for $\theta$ and $\beta$ differ between the two methods. Specifically, compared with AdamW, AdamD is more robust than AdamW when $\theta$ is small, but less robust when $\theta$ is large, as shown in Figures 4(b) and (c).

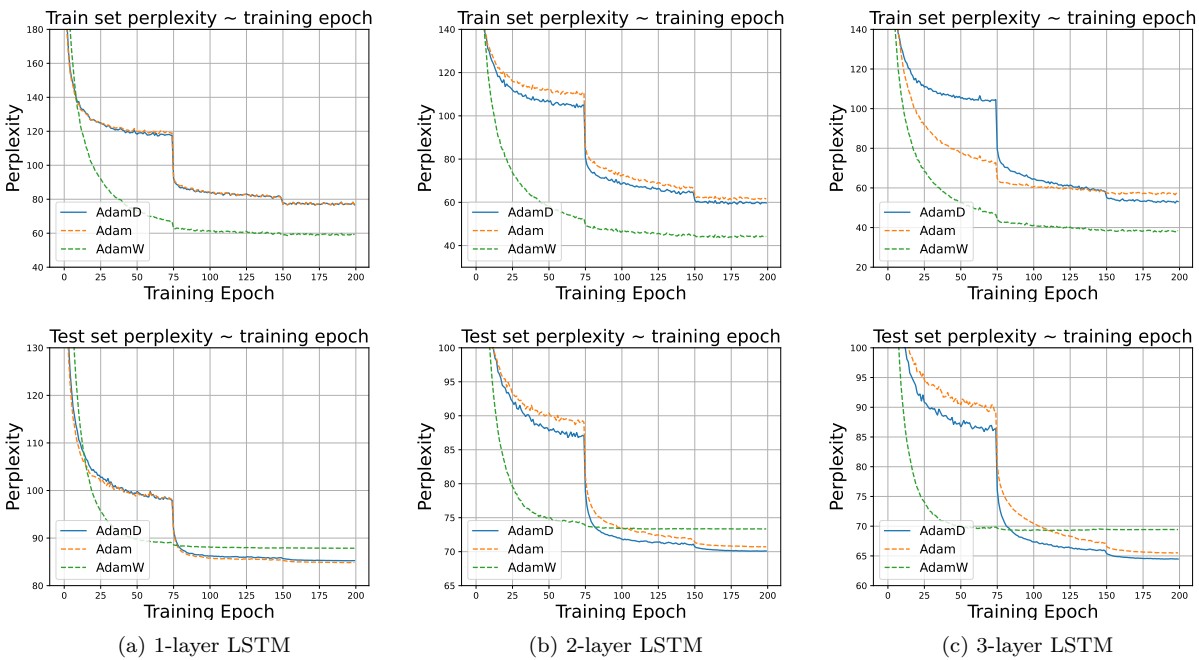

Figure 3: Training and test perplexity (lower is better) of LSTMs on Penn Treebank dataset with stepsize reduced to 0.1 times of the original value at the 75th epoch and 150th epoch.

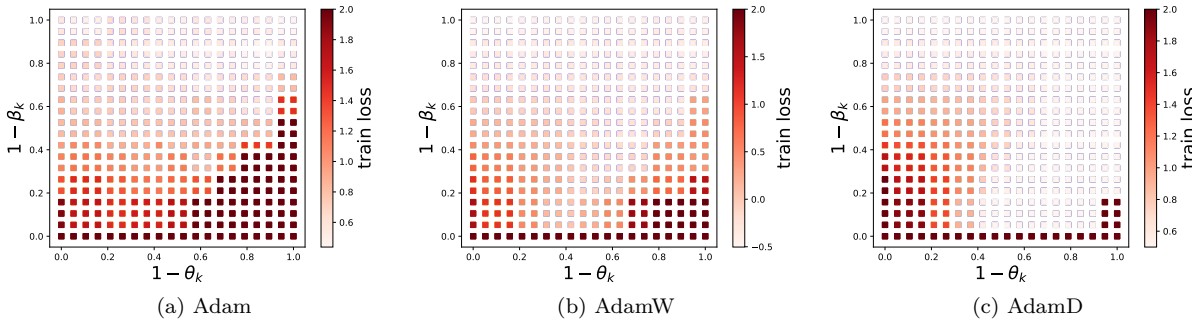

Figure 4: Performance comparison of Adam, AdamW, and AdamD for training a three-layer CNN on the MNIST dataset. Here, the parameters $\{\theta_k\}$ and $\{\beta_k\}$ are set to fixed constants $\theta$ and $\beta$, respectively. The proportion of values below 0.55 is 15.50% for Adam, and 80.25% for both AdamW and AdamD.

### 5.2 Further discussions on the AdamD

#### 5.2.1 Asymptotic approximation to SGD sequence helps generalization

As demonstrated in Lemma 3.7, the term $\|\sigma x_k + m_k\|$ converges to 0 as $k$ tends to infinity. Then as discussed in Lemma 3.8, the sequence $\{y_k\}$ (defined by $y_k := -\sigma^{-1} m_k$) approximately follows the update scheme (14), which asymptotically approximates a SGD method. Together with the fact that $\lim_{k\to\infty} \|x_k - y_k\| = 0$, we can conclude that the sequence $\{x_k\}$ in AdamD is controlled by an SGD sequence $\{y_k\}$ as $k$ goes to infinity. Moreover, the interpolated process of $\{y_k\}$ is a perturbed solution of the differential inclusion (17), i.e.,

$$\frac{\mathrm{d}y}{\mathrm{d}t} \in -(\mathcal{D}_f(y) + \sigma y). \tag{29}$$

On the other hand, in the early stage of the iterations of AdamD, the term $\|\sigma x_k + m_k\|$ is large, and the ratio of $\theta_k$ and $\eta_k$ usually remains nearly unchanged. Then as illustrated in the discussion in Section 4, the sequence $\{(x_k, m_k, v_k)\}$ jointly tracks the trajectories of the differential inclusion

$$\left(\frac{\mathrm{d}x}{\mathrm{d}t}, \frac{\mathrm{d}m}{\mathrm{d}t}, \frac{\mathrm{d}v}{\mathrm{d}t}\right) \in - \begin{bmatrix} (\mathcal{P}_+(v) + \varepsilon)^{-\frac{1}{2}} \odot (m + \sigma x) \\ \tau_1 m - \tau_1 \mathcal{D}_f(x) \\ \tau_2 v - \tau_2 \mathcal{U}(x) \end{bmatrix}. \tag{30}$$

Here $\mathcal{U}(x) := \frac{1}{N} \sum_{i=1}^{N} \{d \odot d : d \in \mathcal{D}_{f_i}(x)\}$. Similar results are also exhibited in (Bianchi et al., 2022; Xiao et al., 2023a). As the differential inclusion (30) imposes preconditioners to the update directions of $\{x_k\}$ based on the second-order moments of the stochastic subgradients, the sequence could quickly converge to a neighborhood of the stationary points.

These theoretical properties explain the fast convergence of AdamD in the early stage of the training and its lower generalization error than Adam with coupled weight decay. Based on the numerical experiments and our theoretical analysis, we believe the ability to asymptotically track an SGD sequence in AdamD helps to explain its superior generalization performance over Adam.

### 5.2.2 Decoupled weight decay is equivalent to quadratic regularization

It is conjectured in (Loshchilov & Hutter, 2019) that the quadratic regularization term contributes to the low generalization error in training neural networks. Moreover, the authors in (Loshchilov & Hutter, 2019) develop AdamW, showing that weight decay is not equivalent to quadratic regularization. As a result, the term $\sigma x_k$ in AdamW is not scaled by the preconditioner $(\sqrt{v_{k+1}} + \varepsilon)^{-1}$. Therefore, AdamW does not have a clear objective function and lacks convergence guarantees in training nonsmooth neural networks.

In our AdamD method, the objective function is exactly the $g(x)$ in (UOP). Hence the weight decay parameter $\sigma$ is exactly the penalty parameter for the quadratic penalty term $\frac{\sigma}{2} \|x\|^2$ in (UOP). More importantly, we provide theoretical guarantees for AdamD in training nonsmooth neural networks. The stationarity of the iterates $\{x_k\}$ is characterized by $\mathcal{D}_f(x_k) + \sigma x_k$, hence has clearer meaning when compared with AdamW.

Furthermore, our numerical experiments demonstrate the superior performance of AdamD, illustrating that employing the quadratic regularization term in (UOP) does not undermine the generalization error. Based on these results, we can conclude that within our framework (AFMDW), the weight decay can be interpreted as the quadratic regularization, which is different from the demonstrations in (Loshchilov & Hutter, 2019) regarding AdamW.

## 6 Conclusion

In this paper, motivated by AdamW, we propose a novel framework (AFMDW) for Adam-family methods with decoupled weight decay. We prove that under mild assumptions with non-diminishing stepsizes $\{\eta_k\}$ and diminishing momentum parameters $\{\theta_k\}$, any cluster point of $\{x_k\}$ is a $\mathcal{D}_g$-stationary point of (UOP). When $\{\theta_k\}$ is also non-diminishing, the sequence $\{x_k\}$ eventually stabilizes around the critical points of the $\mathcal{D}_g$-stationary point of (UOP). Moreover, when employing a single-timescale scheme, any cluster point of $\{x_k\}$ is a $\mathcal{D}_g$-stationary point of (UOP). Compared with AdamW, our proposed framework (AFMDW) enjoys convergence guarantees in training nonsmooth neural networks and yields solutions that have clearer meanings. More importantly, we prove that the decoupled weight decay grants more flexibility of the choices of the parameters $\{\theta_k\}$ and $\{\beta_k\}$ in (AFMDW) than Adam. This fact theoretically illustrates the advantages of the employment of the decoupled weight decay.

As an application of our proposed framework (AFMDW), we develop a novel Adam-family method named Adam with decoupled weight decay (AdamD), and prove its convergence properties under mild conditions. Numerical experiments on image classification and language modeling demonstrate the effectiveness of our proposed method. To conclude, we believe that our work has enriched the theoretical understanding of weight decay and explained its practical utility in the field of deep learning applications.

It is worth mentioning that the formulation of (AFMDW) is slightly different from the widely employed AdamW (Loshchilov & Hutter, 2019). However, it is still challenging to find a Lyapunov function for AdamW to establish its convergence guarantees under general nonsmooth nonconvex settings. We believe that exploring the convergence properties of AdamW in nonsmooth nonconvex optimization is an interesting topic for future research.

**Acknowledgments**

The research of Kim-Chuan Toh is supported by the Ministry of Education, Singapore, under its Academic Research Fund Tier 2 grant (MOE-T2EP20224-0029).

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

## A   Proofs

*Proof of Proposition 3.4.* As illustrated in Assumption 3.3, $d_k \in \mathcal{D}_f^{\delta_k}(x_k)$ and $\{\xi_k\}$ is uniformly bounded. Then it is easy to verify that there exists a constant $\hat{L}$ such that $\|g_k\| = \|d_k + \xi_{k+1}\| \le \hat{L}(1 + \|x_k\|^\nu)$ holds for any $k \ge 0$.

Let the constant $Q$ be defined as

$$Q \ge \max \left\{ \left( \frac{2M_v \hat{L}}{\varepsilon_v \sigma} \right)^{\frac{1}{1-\nu}}, \frac{M_v \|m_0\|}{\varepsilon_v \sigma}, \|x_0\| + 1 \right\}. \tag{31}$$

In the following, for any sequence $\{x_k\}$ generated from (AFMDW), we aim to prove that the set $\{k \ge 0 : \|x_k\| \ge Q\}$ is an empty set by contradiction. Therefore, we assume that the set $\{k \ge 0 : \|x_k\| \ge Q\}$ is non-empty and set $\tau = \inf\{k \ge 0 : \|x_k\| \ge Q\} - 1$. Then from the definition of $\tau$, we have $\|x_{\tau+1}\| \ge Q > \|x_\tau\|$.

On the other hand, from the update scheme (AFMDW), for any $k < \tau$, we have

$$\|m_{k+1}\| \le \max \left\{ m_0, \sup_{0 \le i \le k+1} \|g_i\| \right\} < \max\{\|m_0\|, \hat{L}(1 + Q^\nu)\} \le \frac{\sigma \varepsilon_v}{M_v} Q,$$

where the last inequality follows from the definition of $Q$ and the fact that

$$\hat{L}(1 + Q^\nu) \le 2\hat{L}Q^\nu = \frac{\sigma \varepsilon_v}{M_v} \cdot \frac{2M_v \hat{L}}{\sigma \varepsilon_v} Q^\nu \le \frac{\sigma \varepsilon_v}{M_v} Q^{1-\nu} Q^\nu = \frac{\sigma \varepsilon_v}{M_v} Q.$$

Then it holds that

$$\|x_{\tau+1}\| = \|(1 - \eta_k \sigma H_\tau(v_{\tau+1})) \odot x_\tau - \eta_k H_\tau(v_{\tau+1}) \odot m_{\tau+1}\|$$
$$\le (1 - \eta_k \sigma \varepsilon_v) \|x_\tau\| + \eta_k M_v \|m_{\tau+1}\| < (1 - \eta_k \sigma \varepsilon_v) Q + \eta_k M_v \cdot \frac{\sigma \varepsilon_v}{M_v} Q = Q.$$

But $\|x_{\tau+1}\| < Q$ contradicts to the definition of $\tau$. Thus, the set $\{k \ge 0 : \|x_k\| \ge Q\}$ is empty. Therefore, we have that $\sup_{k \ge 0} \|x_k\| \le Q$ holds almost surely. This completes the proof. □

*Proof of Lemma 3.7.* From Assumption 3.5, there exists a constant $\tilde{\eta} \in (0, 1)$ such that $\max\{|1 - \eta_k \sigma M_v|, |1 - \eta_k \sigma \varepsilon_v|\} \le 1 - \tilde{\eta}$ holds for any $k \ge 0$. Then from the update scheme of $\{x_k\}$ in the framework (AFMDW), almost surely, it holds that

$$\|\sigma x_{k+1} + m_{k+1}\|$$
$$= \|(1 - \eta_k \sigma H(v_{k+1})) \odot (\sigma x_k + m_k) + \theta_k (1 - \eta_k \sigma H(v_{k+1})) \odot (g_k - m_k)\|$$
$$\le \max\{|1 - \eta_k \sigma M_v|, |1 - \eta_k \sigma \varepsilon_v|\}(\|\sigma x_k + m_k\| + \theta_k \|g_k - m_k\|)$$
$$\le (1 - \tilde{\eta}) \|\sigma x_k + m_k\| + 2M_d \theta_k \le (1 - \tilde{\eta})^{k+1} \|\sigma x_0 + m_0\| + 2M_d \sum_{i=0}^{k} (1 - \tilde{\eta})^{k-i} \theta_i \tag{32}$$
$$\le (1 - \tilde{\eta})^{k+1} (\sigma M_x + M_d) + 2M_d \sum_{i=0}^{k} (1 - \tilde{\eta})^{k-i} \theta_i =: \hat{\delta}_{k+1}.$$

Since $\lim_{k \to \infty} \theta_k = 0$, we have $\lim_{k \to \infty} \sum_{i=0}^{k} (1 - \tilde{\eta})^{k-i} \theta_i = 0$. Thus we get $\lim_{k \to \infty} \hat{\delta}_k = 0$, and $\|\sigma x_k + m_k\| \le \hat{\delta}_k$ holds for any $k \ge 0$. This completes the proof. □

*Proof of Lemma 3.8.* As illustrated in Assumption 3.3(2), there exists $\tilde{x}_k \in \mathbb{B}_{\delta_k}(x_k)$ and $\tilde{d}_k \in \mathcal{D}_f(\tilde{x}_k)$ such that $\|d_k - \tilde{d}_k\| \le \delta_k$ and $\lim_{k \to \infty} \delta_k = 0$. Combining with equation (32), it holds that $\|y_k - \tilde{x}_k\| \le$

$\|y_k - x_k\| + \|x_k - \tilde{x}_k\| \leq \frac{\hat{\delta}_k}{\sigma} + \delta_k$. As a result,

$$
\begin{aligned}
\text{dist}\,(d_k + \sigma y_k, \mathcal{D}_g(\tilde{x}_k)) &\leq \left\| d_k + \sigma y_k - (\tilde{d}_k + \sigma \tilde{x}_k) \right\| \\
&\leq \left\| d_k - \tilde{d}_k \right\| + \sigma \left\| y_k - \tilde{x}_k \right\| \leq \delta_k + \sigma(\frac{\hat{\delta}_k}{\sigma} + \delta_k).
\end{aligned}
$$

Since $\tilde{x}_k \in \mathbb{B}_{\delta_k^\star}(y_k)$ and $\text{dist}(d_k + \sigma y_k, \mathcal{D}_g(\tilde{x}_k)) \leq \delta_k^\star$, we get (15). $\qquad \square$

*Proof of Proposition 3.9.* Based on Lemma 2.21, by verifying its conditions, we can prove that the interpolated process of $\{y_k\}$ is a perturbed solution for the differential inclusion (17).

Condition (1) of Lemma 2.21 directly follows from Assumption 3.3(3) and Proposition 2.3, by choosing the stepsizes in (8) as $\{\frac{\theta_k}{\sigma}\}$. Moreover, Lemma 3.8 guarantees the validity of the condition (2) in Lemma 2.21 by noting that $\lim_{k\to\infty} \delta_k^* = 0$. Furthermore, condition (3) of Lemma 2.21 follows from Assumption 3.3(2) and Lemma 3.6. As a result, directly from Lemma 2.21, we can conclude that almost surely, the interpolated process of $\{y_k\}$ is a perturbed trajectory of the differential inclusion (17). $\qquad \square$

*Proof of Theorem 3.10.* From Lemma 3.2 and Proposition 2.18, we can conclude that $g$ is a Lyapunov function for the differential inclusion (17) with the stable set $\{x \in \mathbb{R}^n : 0 \in \mathcal{D}_g(x)\}$. Moreover, Proposition (3.9) illustrates that almost surely, the interpolated process of the sequence $\{y_k\}$ in (13) is a perturbed solution of the differential inclusion (17). As a result, it follows from Theorem 2.22 that any cluster point of $\{y_k\}$ lies in the set $\{x \in \mathbb{R}^n : 0 \in \mathcal{D}_g(x)\}$ and the sequence $\{g(y_k)\}$ converges.

Since $\lim_{k\to\infty} \theta_k = 0$, Lemma 3.7 implies that $\lim_{k\to\infty} \|x_k - y_k\| = 0$ holds almost surely. Then from the continuity of $g$ and the convergence properties of $\{y_k\}$, we can conclude that any cluster point of $\{x_k\}$ lies in the set $\{x \in \mathbb{R}^n : 0 \in \mathcal{D}_g(x)\}$ and the sequence $\{g(x_k)\}$ converges. This completes the proof. $\qquad \square$

*Proof of Lemma 3.12.* From Assumption 3.11, it holds for all $s \geq 0$ and any $i$ satisfying $s \leq i \leq \Lambda(\lambda(s) + T)$ that

$$
\begin{aligned}
\left\| \sum_{k=s}^{i} \theta_k \xi_{k+1} \right\| &\leq \left\| \sum_{k=s}^{N \cdot \lceil \frac{s}{N} \rceil - 1} \theta_k \xi_{k+1} \right\| + \left\| \sum_{k=N \cdot \lceil \frac{s}{N} \rceil}^{N \cdot \lfloor \frac{i}{N} \rfloor - 1} \theta_k \xi_{k+1} \right\| + \left\| \sum_{k=N \cdot \lfloor \frac{i}{N} \rfloor}^{i} \theta_k \xi_{k+1} \right\| \\
&= \left\| \sum_{k=s}^{N \cdot \lceil \frac{s}{N} \rceil - 1} \theta_k \xi_{k+1} \right\| + \left\| \sum_{k=N \cdot \lfloor \frac{i}{N} \rfloor}^{i} \theta_k \xi_{k+1} \right\|.
\end{aligned} \tag{33}
$$

Therefore, for any any $\varepsilon > 0$, choose $\theta_\varepsilon = \frac{\varepsilon}{2NM_\xi}$ guarantees that

$$
\limsup_{s\to+\infty} \sup_{s\leq i\leq\Lambda(\lambda(s)+T)} \left\| \sum_{k=s}^{i} \theta_k \xi_{k+1} \right\| \leq 2NM_\xi \limsup_{s\to+\infty,\ s\leq i\leq\Lambda(\lambda(s)+T)} \theta_i \leq \varepsilon.
$$

This completes the proof. $\qquad \square$

*Proof of Theorem 3.13.* For the update scheme (14), Lemma 3.12 and (Xiao et al., 2023b, Theorem 3.7) illustrate that for any $\varepsilon > 0$, there exists $\hat{\eta}, \hat{\theta}_1, T > 0$ such that whenever $\limsup_{k\geq 0} \eta_k \leq \hat{\eta}$, $\limsup_{k\geq 0} \theta_k \leq \hat{\theta}_1$ and $\{\xi_k\}$ is $(\varepsilon, T, \{\theta_k\})$-controlled, we have

$$
\limsup_{k\to+\infty}\ \text{dist}\,(y_k, \{x \in \mathbb{R}^n : 0 \in \mathcal{D}_g(x)\}) \leq \frac{\varepsilon}{2}. \tag{34}
$$

Then by equation (32) in Lemma 3.7, we have that there exists $\hat{\theta}_2$ such that whenever $\limsup_{k\geq 0} \theta_k \leq \hat{\theta}_2$, $\limsup_{k\to\infty} \|x_k - y_k\| \leq \frac{\varepsilon}{2}$. Therefore, whenever $\limsup_{k\geq 0} \theta_k \leq \min\{\hat{\theta}_1, \hat{\theta}_2\}$, we have that

$$
\limsup_{k\to+\infty}\ \text{dist}\,(x_k, \{x \in \mathbb{R}^n : 0 \in \mathcal{D}_g(x)\}) \leq \limsup_{k\to+\infty}\ \text{dist}\,(y_k, \{x \in \mathbb{R}^n : 0 \in \mathcal{D}_g(x)\}) + \frac{\varepsilon}{2} \leq \varepsilon. \tag{35}
$$

This completes the proof. □

*Proof of Lemma 3.19.* Notice that $f$ is a potential function that admits $\mathcal{D}_f$ as its conservative field, and the function $(x, m, v) \mapsto \left\langle m + \sigma x, (\mathcal{P}_+(v) + \varepsilon)^{-\frac{1}{2}} \odot (m + \sigma x) \right\rangle$ is semi-algebraic and thus definable. Then by the chain rule for conservative field (Bolte & Pauwels, 2021), we can conclude that $h$ is a potential function that admits $\mathcal{D}_h$ as its conservative field. Moreover, as $\mathcal{D}_f$ and $\partial \mathcal{P}_+$ are convex-valued over $\mathbb{R}^n$, it holds that $\mathcal{D}_h$ is convex-valued over $\mathbb{R}^n \times \mathbb{R}^n \times \mathbb{R}^n$. This completes the proof. □

*Proof of Proposition 3.20.* For any trajectory of the differential inclusion (23), there exists $l_f : \mathbb{R}_+ \to \mathbb{R}^n$ and $l_u : \mathbb{R}_+ \to \mathbb{R}^n$ such that $l_f(s) \in \mathcal{D}_f(x(s))$ and $l_u(s) \in \mathcal{U}(x(s), m(s))$ for almost every $s \geq 0$, and

$$(\dot{x}(s), \dot{m}(s), \dot{v}(s)) = \begin{bmatrix} -(\mathcal{P}_+(v(s)) + \varepsilon)^{-\frac{1}{2}} \odot (m(s) + \sigma x(s)) \\ -\tau_1 m(s) + \tau_1 l_f(s) \\ -\tau_2 \mathcal{P}_+(v(s)) + \tau_2 l_u(s) \end{bmatrix}. \tag{36}$$

Then from the formulation of $h$, we have

$$\langle \mathcal{D}_h(x(s), m(s), v(s)), (\dot{x}(s), \dot{m}(s), \dot{v}(s)) \rangle$$

$$\ni - \left\langle l_f(s) + \sigma x(s) + \frac{\sigma}{\tau_1}(\mathcal{P}_+(v(s)) + \varepsilon)^{-\frac{1}{2}} \odot (m(s) + \sigma x(s)), (\mathcal{P}_+(v(s)) + \varepsilon)^{-\frac{1}{2}} \odot (m(s) + \sigma x(s)) \right\rangle$$

$$+ \left\langle (\mathcal{P}_+(v(s)) + \varepsilon)^{-\frac{1}{2}} \odot (m(s) + \sigma x(s)), -m(s) + l_f(s) \right\rangle$$

$$+ \frac{\tau_2}{4\tau_1} \left\langle (m(s) + \sigma x(s))^2 \odot (\mathcal{P}_+(v(s)) + \varepsilon)^{-\frac{3}{2}} \odot \partial \mathcal{P}_+(v(s)), v(s) - l_u(s) \right\rangle$$

$$\leq -\frac{\sigma}{\tau_1} \left\langle (\mathcal{P}_+(v(s)) + \varepsilon)^{-1} \odot (m(s) + \sigma x(s)), m(s) + \sigma x(s) \right\rangle$$

$$- \left\langle (\mathcal{P}_+(v(s)) + \varepsilon)^{-\frac{1}{2}} \odot (m(s) + \sigma x(s)), m(s) + \sigma x(s) \right\rangle$$

$$+ \frac{\tau_2}{4\tau_1} \left\langle (m(s) + \sigma x(s))^2, \mathcal{P}_+(v(s)) \odot (\mathcal{P}_+(v(s)) + \varepsilon)^{-\frac{3}{2}} \right\rangle$$

$$\leq -\frac{\sigma}{\tau_1} \left\langle (\mathcal{P}_+(v(s)) + \varepsilon)^{-1} \odot (m(s) + \sigma x(s)), m(s) + \sigma x(s) \right\rangle$$

$$- \left(1 - \frac{\tau_2}{4\tau_1}\right) \left\langle (\mathcal{P}_+(v(s)) + \varepsilon)^{-\frac{1}{2}} \odot (m(s) + \sigma x(s)), m(s) + \sigma x(s) \right\rangle$$

$$\leq -\frac{\sigma}{\tau_1} \left\langle (\mathcal{P}_+(v(s)) + \varepsilon)^{-1} \odot (m(s) + \sigma x(s)), m(s) + \sigma x(s) \right\rangle.$$

Here the first inequality follows from the fact that $l_u(s) \geq 0$ and $\partial \mathcal{P}_+(v) \odot v = \mathcal{P}_+(v)$. The third inequality follows from the fact that $1 - \frac{\tau_2}{4\tau_1} \geq 0$ in Assumption 3.15(3). Therefore, we can conclude that for any initial point $(x(0), m(0), v(0)) \in \mathbb{R}^n \times \mathbb{R}^n \times \mathbb{R}^n$, it holds for any $t \geq 0$ that

$$h(x(t), m(t), v(t)) - h(x(0), m(0), v(0))$$

$$= \int_0^t \min_{l \in \mathcal{D}_h(x(s), m(s), v(s))} \langle l, (\dot{x}(s), \dot{m}(s), \dot{v}(s)) \rangle \, \mathrm{d}s \tag{37}$$

$$\leq -\frac{\sigma}{\tau_1} \int_0^t \left\langle (\mathcal{P}_+(v(s)) + \varepsilon)^{-1} \odot (m(s) + \sigma x(s)), m(s) + \sigma x(s) \right\rangle \mathrm{d}s.$$

As a result, we can conclude that for any trajectory of the differential inclusion (23), it holds for any $t > 0$ that $h(x(t), m(t), v(t)) \leq h(x(0), m(0), v(0))$.

Now consider the case when $(x(0), m(0), v(0)) \notin \{(x, m, v) \in \mathbb{R}^n \times \mathbb{R}^n \times \mathbb{R}^n : 0 \in \mathcal{D}_g(x), m + \sigma x = 0\}$. Suppose there exists some $T > 0$ such that

$$h(x(T), m(T), v(T)) = h(x(0), m(0), v(0)). \tag{38}$$

Then (37) implies that $m(s) + \sigma x(s) = 0$ holds for almost every $s \in [0, T]$. Therefore, $\dot{m}(s) + \sigma \dot{x}(s) = 0$ and (23) implies that $\dot{x}(s) = 0$ hold for almost every $s \in [0, T]$. As a result, we have

$$0 = \dot{m}(s) \in -\tau_1 m(s) + \tau_1 \mathcal{D}_f(x(s)) = \tau_1 \sigma x(s) + \tau_1 \mathcal{D}_f(x(s))$$

holds for almost every $s \in [0, T]$. Together with the fact that $(x(t), m(t), v(t))$ is absolutely continuous and $\mathcal{D}_f$ is graph-closed and locally bounded, we have that

$$m(0) + \sigma x(0) = 0, \quad 0 \in \mathcal{D}_f(x(0)) + \sigma x(0) = \mathcal{D}_g(x(0)).$$

But the above contradicts the condition that $(x(0), m(0), v(0)) \notin \{(x, m, v) : 0 \in \mathcal{D}_g(x), m + \sigma x = 0\}$. As a result, we can conclude that for any $T > 0$, whenever $(x(0), m(0), v(0)) \notin \{(x, m, v) : 0 \in \mathcal{D}_g(x), m + \sigma x = 0\}$, it holds that

$$h(x(T), m(T), v(T)) < h(x(0), m(0), v(0)).$$

This completes the proof. $\qquad\square$

*Proof of Proposition 3.21.* From the uniform boundedness of $\{m_k\}$, $\{v_k\}$ and $\{g_k\}$ in Lemma 3.16 and Lemma 3.17, and Assumption 3.15(4), we can conclude that $\lim_{k \to \infty} \|m_{k+1} - m_k\| + \|v_{k+1} - v_k\| = 0$. Therefore, there exists a sequence of random variables $\{\tau_k\}$ such that almost surely, $\lim_{k \to \infty} \tau_k = 0$ holds and $\|m_{k+1} - m_k\| + \|v_{k+1} - v_k\| \le \tau_k$.

Then from the formulation of the framework (AFMDW), the sequence $\{(x_k, m_k, v_k)\}$ satisfies the following inclusion

$$(x_{k+1}, m_{k+1}, v_{k+1}) \in (x_k, m_k, v_k) - \eta_k \mathcal{G}^{\tau_k}(x_k, m_k, v_k) - \eta_k(0, -\tau_1 \xi_{k+1}, 0).$$

Then it directly follows from Assumption 3.15(4) and Proposition 2.3 that

$$\lim_{s \to \infty} \sup_{s \le i \le \Lambda(\lambda_s + T)} \left\| \sum_{k=s}^{i} \eta_k(0, \tau_1 \xi_{k+1}, 0) \right\| = 0.$$

Therefore, we can conclude that the conditions (1) and (2) in Lemma 2.21 hold. Moreover, condition (3) in Lemma 2.21 directly follows from Assumption 3.15(1), Lemma 3.16 and Lemma 3.17. Therefore, from Lemma 2.21, we can conclude that the interpolated process of $\{(x_k, m_k, v_k)\}$ is a perturbed solution for the differential inclusion (23). This completes the proof. $\qquad\square$

*Proof of Theorem 3.22.* From Proposition 3.21, we can conclude that the interpolated process of $\{(x_k, m_k, v_k)\}$ is a perturbed solution for the differential inclusion (23). Moreover, Proposition 3.20 illustrates that $h$ is a Lyapunov function for the differential inclusion (23) with stable set $\{(x, m, v) \in \mathbb{R}^n \times \mathbb{R}^n \times \mathbb{R}^n : 0 \in \mathcal{D}_g(x), m + \sigma x = 0\}$. Then we can conclude that any cluster point of $\{(x_k, m_k, v_k)\}$ lies in the set $\{(x, m, v) \in \mathbb{R}^n \times \mathbb{R}^n \times \mathbb{R}^n : 0 \in \mathcal{D}_g(x), m + \sigma x = 0\}$, and the sequence $\{h(x_k, m_k, v_k)\}$ converges.

Thus, we can conclude that any cluster point of $\{x_k\}$ lies in the set $\{x \in \mathbb{R}^n : 0 \in \mathcal{D}_g(x)\}$, and any cluster point of $\{(x_k, m_k)\}$ lies in $\{(x, m) \in \mathbb{R}^n \times \mathbb{R}^n : \sigma x + m = 0\}$. As a result, noting that $\{\sigma x_k + m_k\}$ is bounded in $\mathbb{R}^n$, it holds that $\lim_{k \to \infty} \|\sigma x_k + m_k\| = 0$. Furthermore, since

$$\lim_{k \to \infty} |h(x_k, m_k, v_k) - g(x_k)| \le \lim_{k \to \infty} \frac{1}{2\tau_1 \sqrt{\varepsilon}} \|\sigma x_k + m_k\|^2 = 0,$$

we can deduce that the sequence $\{g(x_k)\}$ converges. This completes the proof. $\qquad\square$

*Proof of Theorem 4.2.* We first verify the validity of Assumption 3.1. The definability of $f_i$ and $\mathcal{D}_{f_i}$ implies the definability of $f$ and $\mathcal{D}_f$, hence from (Bolte & Pauwels, 2021, Theorem 5), $f$ is path-differentiable and the set $\{f(x) : 0 \in \mathcal{D}_f(x)\}$ is a finite subset of $\mathbb{R}$. Additionally, Assumption 4.1(2) ensures the validity of Assumption 3.1(2). This verifies the validity of Assumption 3.1.

Moreover, let $\{\mathcal{F}_k\}$ be a sequence of $\sigma$-algebras generated by $\{x_j, d_j, m_j : j \le k\}$, $d_k = \mathbb{E}[g_k | \mathcal{F}_k]$ and $\xi_{k+1} = g_k - d_k$. Then we can conclude that $d_k \in \mathcal{D}_f(x_k)$ and $\mathbb{E}[\xi_{k+1} | \mathcal{F}_k] = 0$. Moreover, Assumption 4.1(2) illustrates that there exists a constant $M_f$ such that $\sup_{i \in [N], x \in \mathbb{R}^n} \|\mathcal{D}_f(x)\| \le M_f$. Thus we can conclude that $\sup_{k \ge 0} \|g_k\| \le M_f$ and $\sup_{k \ge 0} \|d_k\| \le M_f$ hold almost surely. Then $\sup_{k \ge 0} \|\xi_{k+1}\| \le 2M_f$ holds almost surely. This verifies the validity of Assumption 3.3(3).

Furthermore, from the update scheme in Step 5 of Algorithm 1, we can conclude that $\sup_{k\geq 0} \|v_k\| \leq \sup_{k\geq 0} \|g_k^2\| \leq M_f^2$. This illustrates that Assumption 3.3(1) holds with $\varepsilon_v = \frac{1}{M_f+\varepsilon}$ and $M_v = \frac{1}{\varepsilon}$. Therefore, from Theorem 3.10, we can conclude that any cluster point of the sequence $\{x_k\}$ is a $\mathcal{D}_g$-stationary point of $g$, and the sequence $\{g(x_k)\}$ converges. This completes the proof. □

## B   Experiments details

In this section, we provide additional experimental results to further illustrate the performance and robustness of our methods. The following tables report the outcomes on various tasks under different hyperparameter settings.

### B.1   Image Classification:

Tables 4–7 present the performance of ResNet34 and DenseNet121 on CIFAR10 and CIFAR100 datasets. In these experiments, we vary the learning rate (LR) and weight decay parameter for the optimizers AdamD, Adam, and AdamW. The best performance for each fixed weight decay parameter $\sigma$ is highlighted in bold.

| $\eta$ | $\sigma$: 5e-03 | | | $\sigma$: 1e-03 | | | $\sigma$: 5e-04 | | | $\sigma$: 1e-04 | | |
|---|---|---|---|---|---|---|---|---|---|---|---|---|
| | AdamD | Adam | AdamW | AdamD | Adam | AdamW | AdamD | Adam | AdamW | AdamD | Adam | AdamW |
| 1e-01 | 77.05 | 17.84 | 92.01 | 91.23 | 59.13 | 93.39 | 93.67 | 62.71 | 93.19 | 94.41 | 65.09 | 92.22 |
| 5e-02 | 85.03 | 55.27 | 93.28 | 93.68 | 59.33 | 94.31 | **95.22** | 63.57 | 93.93 | **94.96** | 76.80 | 93.72 |
| 1e-02 | 94.08 | 62.33 | 94.54 | **95.10** | 79.67 | 94.16 | 94.54 | 81.43 | 94.22 | 94.28 | 88.96 | 94.09 |
| 5e-03 | **95.13** | 73.20 | 94.28 | 94.67 | 84.46 | 94.19 | 94.53 | 86.81 | 94.33 | 93.69 | 91.47 | 94.27 |
| 1e-03 | 94.57 | 89.81 | 94.40 | 93.05 | 92.48 | 94.34 | 92.54 | 93.31 | 94.18 | 92.12 | 94.06 | 94.56 |
| 5e-04 | 93.42 | 92.25 | 94.19 | 91.60 | 93.46 | 94.40 | 91.24 | 93.83 | 94.26 | 90.67 | 94.29 | 94.39 |
| 1e-04 | 87.64 | 94.09 | 94.09 | 87.22 | 94.04 | 93.86 | 87.47 | 94.10 | 94.58 | 86.62 | 93.89 | 93.71 |
| 5e-05 | 85.91 | 93.63 | 93.11 | 85.90 | 93.64 | 92.65 | 84.68 | 93.23 | 92.86 | 85.03 | 92.78 | 92.96 |

Table 4: ResNet34 on CIFAR10.

| $\eta$ | $\sigma$: 5e-03 | | | $\sigma$: 1e-03 | | | $\sigma$: 5e-04 | | | $\sigma$: 1e-04 | | |
|---|---|---|---|---|---|---|---|---|---|---|---|---|
| | AdamD | Adam | AdamW | AdamD | Adam | AdamW | AdamD | Adam | AdamW | AdamD | Adam | AdamW |
| 1e-01 | 68.39 | 46.88 | 89.47 | 90.13 | 59.13 | 93.55 | 93.80 | 59.58 | 93.26 | 94.52 | 67.28 | 93.24 |
| 5e-02 | 81.32 | 51.89 | 93.21 | 94.02 | 59.33 | 93.96 | 95.47 | 65.89 | 93.72 | **94.73** | 76.94 | 93.11 |
| 1e-02 | 94.09 | 66.60 | 94.21 | **95.68** | 79.67 | 93.94 | **95.25** | 84.27 | 93.51 | 94.48 | 88.40 | 93.58 |
| 5e-03 | **95.22** | 71.17 | 94.44 | 95.04 | 84.46 | 94.01 | 94.75 | 87.19 | 93.68 | 93.99 | 90.51 | 94.07 |
| 1e-03 | 95.04 | 89.91 | 94.29 | 92.65 | 92.48 | 94.54 | 91.94 | 93.57 | 94.32 | 91.55 | 94.10 | 94.49 |
| 5e-04 | 93.57 | 92.37 | 94.69 | 90.88 | 93.46 | 94.83 | 90.62 | 94.11 | 94.41 | 90.48 | 94.32 | 94.38 |
| 1e-04 | 86.08 | 94.14 | 94.16 | 85.16 | 94.10 | 93.86 | 84.60 | 94.10 | 93.93 | 84.53 | 93.88 | 93.58 |
| 5e-05 | 81.22 | 93.64 | 92.47 | 80.96 | 93.64 | 92.29 | 81.53 | 92.35 | 91.82 | 81.34 | 92.38 | 91.96 |

Table 5: DenseNet121 on CIFAR10

| $\eta$ | $\sigma$: 5e-03 | | | $\sigma$: 1e-03 | | | $\sigma$: 5e-04 | | | $\sigma$: 1e-04 | | |
|---|---|---|---|---|---|---|---|---|---|---|---|---|
| | AdamD | Adam | AdamW | AdamD | Adam | AdamW | AdamD | Adam | AdamW | AdamD | Adam | AdamW |
| 1e-01 | 44.78 | 1.71 | 55.48 | 71.37 | 14.12 | 70.27 | 75.40 | 18.67 | 71.21 | 73.80 | 28.14 | 69.27 |
| 5e-02 | 60.58 | 8.19 | 72.09 | 74.71 | 21.98 | 72.39 | **77.22** | 25.05 | 69.82 | 72.79 | 37.42 | 69.70 |
| 1e-02 | 76.64 | 19.20 | 70.68 | **77.26** | 41.02 | 70.65 | 76.16 | 42.35 | 67.47 | **74.74** | 56.91 | 68.50 |
| 5e-03 | **77.74** | 30.09 | 68.44 | 76.47 | 54.23 | 67.86 | 75.73 | 58.16 | 67.71 | 74.11 | 67.05 | 67.10 |
| 1e-03 | 75.11 | 60.86 | 72.69 | 72.18 | 71.26 | 72.73 | 70.86 | 73.25 | 72.33 | 70.01 | 74.55 | 72.83 |
| 5e-04 | 73.15 | 71.67 | 73.99 | 68.11 | 73.71 | 73.82 | 66.86 | 74.36 | 73.79 | 67.31 | 74.65 | 73.78 |
| 1e-04 | 60.71 | 74.31 | 73.18 | 59.29 | 75.00 | 72.99 | 58.83 | 75.42 | 73.09 | 58.85 | 74.53 | 73.17 |
| 5e-05 | 55.21 | 73.84 | 71.11 | 54.67 | 73.31 | 70.77 | 53.93 | 72.54 | 70.47 | 54.45 | 71.32 | 70.90 |

Table 6: ResNet34 on CIFAR100

| $\eta$ | $\sigma$: 5e-03 | | | $\sigma$: 1e-03 | | | $\sigma$: 5e-04 | | | $\sigma$: 1e-04 | | |
|---|---|---|---|---|---|---|---|---|---|---|---|---|
| | AdamD | Adam | AdamW | AdamD | Adam | AdamW | AdamD | Adam | AdamW | AdamD | Adam | AdamW |
| 1e-01 | 34.09 | 9.07 | 48.00 | 68.93 | 17.65 | 68.67 | 75.90 | 26.13 | 64.31 | 75.66 | 34.29 | 61.80 |
| 5e-02 | 51.52 | 15.52 | 69.43 | 77.81 | 22.10 | 69.12 | **78.93** | 30.30 | 65.41 | **77.05** | 39.90 | 65.60 |
| 1e-02 | 78.38 | 23.09 | 70.97 | **79.28** | 49.81 | 71.15 | 77.65 | 48.65 | 70.57 | 76.06 | 64.45 | 70.63 |
| 5e-03 | **79.42** | 28.61 | 71.90 | 77.54 | 52.05 | 70.82 | 76.42 | 59.47 | 70.25 | 74.42 | 69.14 | 70.01 |
| 1e-03 | 76.27 | 64.95 | 75.08 | 71.94 | 72.93 | 74.79 | 70.27 | 73.85 | 75.42 | 70.10 | 74.86 | 75.11 |
| 5e-04 | 73.77 | 73.25 | 76.20 | 67.19 | 75.12 | 76.34 | 65.47 | 76.28 | 76.10 | 67.17 | 75.61 | 75.98 |
| 1e-04 | 54.30 | 74.29 | 73.82 | 52.55 | 74.44 | 73.23 | 51.84 | 73.97 | 74.01 | 52.09 | 73.62 | 73.46 |
| 5e-05 | 42.13 | 71.99 | 69.23 | 42.55 | 70.03 | 68.29 | 42.10 | 69.56 | 68.23 | 42.47 | 69.40 | 68.12 |

Table 7: DenseNet121 on CIFAR100

Additionally, Tables 8 and 9 present experiments on CIFAR100 using the AdamW-D variant, which employs a diminishing momentum parameter $\theta_s$ for AdamW. The results indicate that using a diminishing momentum parameter in AdamW is effective, achieving performance comparable to that of the original AdamW.

| $\eta$ | $\sigma$: 5e-03 | | $\sigma$: 1e-03 | | $\sigma$: 5e-04 | | $\sigma$: 1e-04 | |
|---|---|---|---|---|---|---|---|---|
| | AdamW | AdamW-D | AdamW | AdamW-D | AdamW | AdamW-D | AdamW | AdamW-D |
| 1e-01 | 55.48 | 63.90 | 70.27 | 70.03 | 71.21 | 70.90 | 69.27 | 66.50 |
| 5e-02 | 72.09 | 70.24 | 72.39 | 69.95 | 69.82 | 70.06 | 69.70 | 70.72 |
| 1e-02 | 70.68 | 72.02 | 70.65 | 70.46 | 67.47 | 70.10 | 68.50 | 69.71 |
| 5e-03 | 68.44 | 69.77 | 67.86 | 69.72 | 67.71 | 68.43 | 67.10 | 67.23 |
| 1e-03 | 72.69 | 73.19 | 72.73 | 72.58 | 72.33 | 72.83 | 72.83 | 72.81 |
| 5e-04 | 73.99 | **74.26** | 73.82 | **73.99** | 73.79 | **73.85** | **73.78** | 73.61 |
| 1e-04 | 73.18 | 73.38 | 72.99 | 73.08 | 73.09 | 73.49 | 73.17 | 73.66 |
| 5e-05 | 71.11 | 70.51 | 70.77 | 70.47 | 70.47 | 70.20 | 70.90 | 70.71 |

Table 8: ResNet34 on CIFAR100: diminishing $\theta_s$ for AdamW-D

| $\eta$ | $\sigma$: 5e-03 | | $\sigma$: 1e-03 | | $\sigma$: 5e-04 | | $\sigma$: 1e-04 | |
|---|---|---|---|---|---|---|---|---|
| | AdamW | AdamW-D | AdamW | AdamW-D | AdamW | AdamW-D | AdamW | AdamW-D |
| 1e-01 | 48.00 | 55.29 | 68.67 | 68.57 | 64.31 | 63.80 | 61.80 | 65.40 |
| 5e-02 | 69.43 | 67.20 | 69.12 | 66.55 | 65.41 | 67.23 | 65.60 | 68.11 |
| 1e-02 | 70.97 | 71.04 | 71.15 | 70.92 | 70.57 | 68.98 | 70.63 | 70.48 |
| 5e-03 | 71.90 | 72.20 | 70.82 | 71.29 | 70.25 | 70.67 | 70.01 | 70.80 |
| 1e-03 | 75.08 | 75.06 | 74.79 | 74.95 | 75.42 | 75.73 | 75.11 | 74.74 |
| 5e-04 | 76.20 | 76.32 | **76.34** | 75.79 | **76.10** | 75.85 | 75.98 | **76.45** |
| 1e-04 | 73.82 | **74.02** | 73.23 | 73.77 | 74.01 | 73.18 | 73.46 | 73.23 |
| 5e-05 | 69.23 | 68.70 | 68.29 | 69.20 | 68.23 | 69.38 | 68.12 | 68.55 |

Table 9: DenseNet121 on CIFAR100: diminishing $\theta_s$ for AdamW-D

## B.2 Language Modeling with LSTMs:

Tables 10–12 summarize the performance of 1-layer, 2-layer, and 3-layer LSTM models under various learning rate and weight decay configurations. Figure 5 illustrates the training and test perplexity curves for LSTM models with different numbers of layers, highlighting the convergence behavior and generalization performance of the optimizers. In Figure 5, the weight decay parameter is fixed at $\sigma = 10^{-5}$, and the learning rates are tuned among $\{10^{-2}, 5 \times 10^{-3}, 10^{-3}\}$ based on observations from Tables 10 and 12.

From Figure 5, we observe that AdamD consistently outperforms Adam and AdamW in terms of test performance for models with more than two layers. Moreover, the performance gap between AdamD and the

other methods widens as the number of layers increases. Note that for models with more than five layers, overfitting may occur, as indicated by an increase in test perplexity.

| $\eta$ | $\sigma$: 1e-04 | | | $\sigma$: 1e-05 | | | $\sigma$: 1e-06 | | |
|---|---|---|---|---|---|---|---|---|---|
| | AdamD | Adam | AdamW | AdamD | Adam | AdamW | AdamD | Adam | AdamW |
| 1e-01 | 166.1 | 172.8 | 560.6 | 582.9 | 88.2 | 582.9 | 128.5 | 173.0 | 572.8 |
| 5e-02 | 165.2 | 167.6 | 361.9 | 381.3 | 85.3 | 381.3 | 95.1 | 95.8 | 372.3 |
| 1e-02 | 163.2 | 165.2 | 139.1 | 85.8 | 85.6 | 139.7 | 92.0 | 90.9 | 138.9 |
| 5e-03 | 163.5 | 163.9 | 102.9 | 85.2 | **85.0** | 103.2 | 95.3 | 85.8 | 103.3 |
| 1e-03 | 171.1 | 165.9 | **87.8** | 90.1 | 89.7 | 87.8 | 85.4 | **85.1** | 87.8 |
| 5e-04 | 182.6 | 169.3 | 88.7 | 96.2 | 94.8 | 88.7 | 88.5 | 88.0 | 88.7 |

Table 10: 1-layer LSTM

| $\eta$ | $\sigma$: 1e-04 | | | $\sigma$: 1e-05 | | | $\sigma$: 1e-06 | | |
|---|---|---|---|---|---|---|---|---|---|
| | AdamD | Adam | AdamW | AdamD | Adam | AdamW | AdamD | Adam | AdamW |
| 1e-01 | 602.1 | 158.5 | 602.1 | 623.9 | 75.0 | 623.9 | 92.4 | 93.3 | 623.8 |
| 5e-02 | 489.3 | 197.1 | 502.5 | 507.2 | 71.8 | 507.2 | 70.2 | 71.2 | 446.5 |
| 1e-02 | 145.1 | 147.2 | 93.0 | 70.9 | 71.4 | 93.3 | 68.4 | 68.1 | 93.5 |
| 5e-03 | 146.8 | 147.0 | 83.6 | **70.0** | 70.7 | 82.9 | 83.4 | **67.9** | 83.2 |
| 1e-03 | 154.4 | 147.5 | **73.2** | 74.9 | 74.5 | 73.3 | 71.2 | 71.0 | 73.3 |
| 5e-04 | 170.5 | 151.2 | 73.3 | 79.9 | 78.5 | 73.3 | 72.4 | 72.4 | 73.3 |

Table 11: 2-layer LSTM

| $\eta$ | $\sigma$: 1e-04 | | | $\sigma$: 1e-05 | | | $\sigma$: 1e-06 | | |
|---|---|---|---|---|---|---|---|---|---|
| | AdamD | Adam | AdamW | AdamD | Adam | AdamW | AdamD | Adam | AdamW |
| 1e-01 | 141.7 | 154.6 | 642.1 | 638.9 | 72.9 | 638.9 | 69.8 | 69.7 | 638.9 |
| 5e-02 | 138.4 | 145.9 | 587.1 | 573.5 | 67.9 | 573.5 | 65.2 | 65.0 | 591.4 |
| 1e-02 | 135.0 | 191.2 | 106.0 | 64.9 | 66.7 | 99.4 | **64.9** | 65.2 | 99.1 |
| 5e-03 | 134.5 | 138.8 | 81.0 | **64.4** | 65.4 | 80.9 | 73.5 | 65.9 | 81.1 |
| 1e-03 | 144.7 | 137.8 | **69.0** | 69.6 | 67.1 | 69.2 | 67.3 | 66.6 | 69.2 |
| 5e-04 | 164.1 | 139.8 | 69.2 | 69.2 | 70.7 | 69.2 | 68.3 | 68.1 | 69.2 |

Table 12: 3-layer LSTM

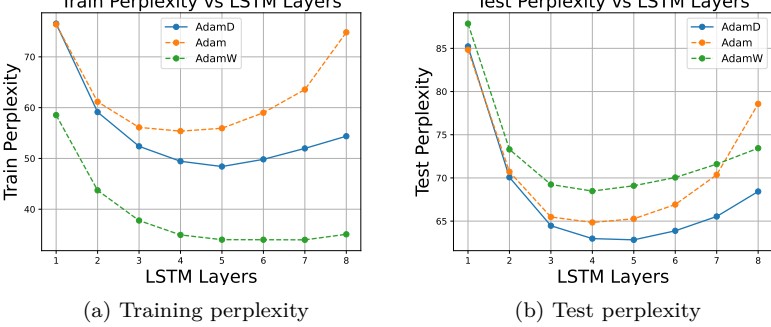

(a) Training perplexity      (b) Test perplexity

Figure 5: Comparison of training and test perplexity curves across LSTM models with different numbers of layers and three optimizers.

