# OpenReview forum: "Adam-family Methods with Decoupled Weight Decay in Deep Learning"
_TMLR — Accepted by TMLR_

### Review · Reviewer_tSGb · 2025-02-19

**Summary Of Contributions:**

The paper analyzes a family of Adam-type methods in the nonsmooth, nonconvex setting. A global asymptotic convergence and stability  result is shown for a general notion of the subdifferential. Finally, one specific member of this family of methods is proposed and analyzed in experiments.

**Audience:**

Yes

**Claims And Evidence:**

Yes

**Requested Changes:**

Please add discussions and references related to the prior works mentioned in Weaknesses. Also, a comparison to the this paper https://openreview.net/forum?id=ZPQhzTSWA7 is missing, which also analyses Adam.

**Changes for experiments:**

1) I understand that the effect of weight-decay is different for AdamW vs AdamD due to the fact that the term $\sigma x_k$ is multiplied by $H_k$ or not. But in consequence, why is the weight decay value $\sigma$ fixed, while the range of learning rates $\eta$ is different for Adam(W) and AdamD by one order? Wouldn't it make more sense to compensate by changing the value of $\sigma$, while doing exactly the same grid search over $\eta$ for all methods (as $\eta$ also impacts the gradient term)?

2) Please state for each method and problem setting the tuned value of $\eta$. Possible performance differences might be due to the tuning procedure, and it is currently incomprehensible for the reader as these details are missing. In general, it is also helpful to see what is the normal variation of a method only due to tuning $\eta$. For example, one could show the K best step sizes/runs for each method in the appendix.

3) Was the choice of $\theta_s$ used for AdamD also tried for Adam(W)? If yes, how does it compare?

**Minor:**

* There are many citations that should use "\citet" instead of "\citep", for example last line of page 1. Or "Kingma and Ba (Kingma & Ba 2015)" on page 1.
* The paper cited as "Loshchilov & Hutter 2017" was published in 2019 at ICLR. As these two authors have another widely known paper from 2017, this can lead to confusion (so it should be "Loshchilov & Hutter 2019" instead).
* Page 26: "Proof ofof" should be "Proof of".

**Strengths And Weaknesses:**

Strengths:
This paper analyzes Adam-type methods in a very general nonsmooth nonconvex setting. In particular, the analysis is valid for a subdifferential that includes the "gradients" that are usually computed from automatic differentiation. This keeps the analysis close to what is actually done in practice. Another strength is that the analysis in fact includes many different methods (see Table 2) within a unifying proof technique.

Weaknesses:
* The convergence results are asymptotic. No non-asymptotic guarantees or convergence rates are provided.

* A TMLR paper by Zhuang et al., 2022 (https://openreview.net/forum?id=IKhEPWGdwK) shows how AdamW can be seen as an approximation of proximal Adam (that is, handling the regularization with proximal operators). This important reference is missing in the discussion. For example, on page 3 bottom, it says: "AdamW lacking a clearly defined objective function to minimize". In the light of Zhuang et al, 2022, this is not entirely true: AdamW can be seen as minimizing a regularized objective with a proximal method (up to a first-order Taylor approximations).

* The proposed method AdamD seems to have been proposed almost identically in this (peer-reviewed) blog post: https://iclr-blogposts.github.io/2023/blog/2023/adamw/. If one looks at the method called AdamP therein, then doing the same first-order Taylor approximation as mentioned above on the coefficients, one obtains AdamD. To be clear, the contributions of this submission still go beyond the blog post as (i) a theoretical analysis of AdamD is provided in a very general setting and (ii) the numerical evaluation in this paper is much more detailed and extensive than the one in the blog post. However, a discussion and reference to these previous works should be included.

* Several details on the experimental section are unclear or not provided (see *Requested Changes*)

* Page 2: "Given the fact that non-diminishing step-sizes are widely employed in computational frameworks, ...". Are non-diminishing step sizes really the standard practice for training? To my knowledge, a common choice is to use step size schedulers with some form of annealing in the end (cosine, WSD, linear decay); step sizes of the form 1/sqrt(t) are also widely established in theory and practice.

---

> ### Author Response · Authors · 2025-03-18
> **Authors’ response to Reviewer tSGb**
>
> We thank the reviewer tSGb for the valuable comments on the experiments part! Please find our responses to the questions below. The comprehensive revisions on experiments part (marked in blue) can be found in the revised text (Section 5 and Appendix B).
>
> **Q1：** I understand that the effect of weight-decay is different for AdamW vs AdamD due to the fact that the term $\sigma x_k$ is multiplied by $H_k$ or not. But in consequence, why is the weight decay value $\sigma$ fixed, while the range of learning rates $\eta$ is different for Adam(W) and AdamD by one order? Wouldn't it make more sense to compensate by changing the value of $\sigma$, while doing exactly the same grid search over $\eta$ for all methods (as $\eta$ also impacts the gradient term)?
>
> ***A1：*** We appreciate this insightful comment. In response, we have revised our hyperparameter settings in Section 5. Specifically, we now consider weight decay values in $\sigma \in {5\times10^{-3},, 10^{-3},, 5\times10^{-4},, 10^{-4}}$. By fixing $\sigma$ first, we ensure that all methods are solving the same minimization problem. With $\sigma$ fixed, we perform a grid search over the learning rate $\eta$ for AdamD, Adam, and AdamW using $\eta\in\{5\times10^{-5},10^{-4},5\times10^{-4},10^{-3},5\times10^{-3},10^{-2},5\times10^{-2},10^{-1}\}$. The tuned step sizes for Figures 1 and 3 are summarized in Table 3.
>
> **Q2：** Please state for each method and problem setting the tuned value of $\eta$. Possible performance differences might be due to the tuning procedure, and it is currently incomprehensible for the reader as these details are missing. In general, it is also helpful to see what is the normal variation of a method only due to tuning $\eta$. For example, one could show the K best step sizes/runs for each method in the appendix.
>
> ***A2：*** We have applied the same $\theta_s$ used for AdamD to AdamW. Please refer to Tables 7 and 8. The results indicate that using a diminishing momentum parameter in AdamW is effective, achieving performance comparable to that of the original AdamW.
>
> **Q3：** Was the choice of $\theta_s$ used for AdamD also tried for Adam(W)? If yes, how does it compare?
>
> ***A3：*** We have tried the same $\theta_s$ used for AdamD for AdamW. See Table 7 and 8. The results indicate that using a diminishing momentum parameter in AdamW is effective, achieving performance comparable to that of the original AdamW.
>
> **Minor：** Using "\citet" instead of "\citep"; citing "Loshchilov & Hutter" ICLR version; typo: "Proof ofof"
>
> ***Ans：*** We have corrected these issues throughout the manuscript.

---

> > ### Comment · Reviewer_tSGb · 2025-03-19
> > **Response to revised paper**
> >
> > Dear authors,
> >
> > thanks for providing additional details in the experiment section according to the suggestion from the review.
> >
> > Can you comment on the related works:
> >
> > * https://openreview.net/forum?id=IKhEPWGdwK
> > * https://iclr-blogposts.github.io/2023/blog/2023/adamw/
> > * https://openreview.net/forum?id=ZPQhzTSWA7
> >
> > I could not find a discussion in the revised PDF, and as my initial review mentioned, it should be clarified that (a) the proposed method AdamD has been proposed previously almost identically, and (b) it is known that AdamW is (approximately) a proximal version of Adam (thus, the relationship of AdamW to the regularized objective is much better understood as what is stated in the paper currently).

---

> > > ### Author Response · Authors · 2025-03-19
> > > **Authors’ response to Reviewer tSGb**
> > >
> > > We thank the reviewer for the valuable comments on the related references! Please find our responses to the questions below. The  revisions part (marked in blue) can be found in the Introduction Section.
> > >
> > > Please find our responses to the questions below. Here are some clarifications on the differences between our AdamD algorithms and the results in [Zhuang et.al. 2022](https://openreview.net/forum?id=IKhEPWGdwK), [Schaipp 2023](https://iclr-blogposts.github.io/2023/blog/2023/adamw/) and [Défossez et.al. 2022](https://openreview.net/forum?id=ZPQhzTSWA7).
> > >
> > > * For the proximal version of the adaptive method duscussed in [Zhuang et.al. 2022](https://openreview.net/forum?id=IKhEPWGdwK) and [Schaipp 2023](https://iclr-blogposts.github.io/2023/blog/2023/adamw/), the $\{x_k\}$ is updated by performing the proximal mapping to the direction $\frac{m_{k+1}}{\sqrt{v_{k+1}}+ \varepsilon}$, hence resulting the update scheme
> > >   $$
> > >   x_{k+1} = (1+\sigma \eta_k)^{-1} \left(x_k - \eta_k \frac{m_{k+1}}{\sqrt{v_{k+1}}+ \varepsilon}\right)
> > >   $$
> > >   As discussed in Page 4 of their paper, they demonstrate that their proposed schemes resembles the AdamW method.
> > >
> > >   However, the update scheme of $\{x_k\}$ in AdamD is
> > >   $$
> > >   x_{k+1} = x_k - \eta_k \frac{m_{k+1} + \sigma x_k}{\sqrt{v_{k+1}}+ \varepsilon},
> > >   $$
> > >   which is significantly different from the update scheme of AdamW. Therefore, we believe the adaptive methods discussed in [Zhuang et.al. 2022](https://openreview.net/forum?id=IKhEPWGdwK) and [Schaipp 2023](https://iclr-blogposts.github.io/2023/blog/2023/adamw/) significantly differ from our proposed AdamD method, as illustrated in the comparison of Algorithm 1 and AdamW on Page 17 of our paper. Therefore, we have modified the first paragraph on Page 4 of our paper to highlight that these methods are proximal variants of AdamW.
> > >
> > >
> > > * For the adaptive method discussed in Equation (1)-(3) of [Défossez et.al. 2022](https://openreview.net/forum?id=ZPQhzTSWA7), it  is exactly the original Adam method proposed by [Kingma & Ba 2015](https://arxiv.org/abs/1412.6980). The original Adam method does not involve the decoupled weight decay, in the sense that the momentum terms $\{m_k\}$ are updated by $m_{k+1} = (1-\theta_k)m_k + \theta_k (g_k + \sigma x_k)$, while the $\{x_k\}$ is updated by
> > >   $$
> > >   x_{k+1} = x_k - \eta_k \frac{m_{k+1}}{\sqrt{v_{k+1}}+ \varepsilon}.
> > >   $$
> > >   Compared with the Algorithm 1 in our paper, we can conclude that our proposed AdamD method differs from the adaptive method discussed in [Défossez et.al. 2022](https://openreview.net/forum?id=ZPQhzTSWA7) whenever $\sigma > 0$. We have modified the last paragraph on Page 1 to clarify that  [Défossez et.al. 2022](https://openreview.net/forum?id=ZPQhzTSWA7) only discusses the convergence of original Adam method.
> > >
> > > **References:**
> > >
> > > (Défossez et.al. 2022) A simple convergence proof of adam and adagrad. TMLR.
> > >
> > > (Schaipp, 2023)  Decay no more. ICLR Blogposts.
> > >
> > > (Zhuang et. al. 2022) Understanding adamw through proximal methods and scale-freeness. TMLR.

---

> > > > ### Comment · Reviewer_tSGb · 2025-03-19
> > > > **Response**
> > > >
> > > > On the first bullet point:
> > > >
> > > > Yes, ProxAdam from Zhuang et al. 2022 is very different to AdamD, but the method called AdamP in Schaipp 2023 (introduced rather late in the blog post) not so much. It is given by (denoting $D_k = \epsilon + \sqrt{v_{k+1}}$)
> > > >
> > > > $$ x_{k+1} = \frac{1}{1+\eta_k\sigma D_k^{-1}}  (x_k -  \eta_k D_k^{-1}m_k). $$
> > > >
> > > > Now, Taylor approximation (at $\eta_k  D_k^{-1}$ close to zero) for the coefficients in front of $x_k$ and $-m_k$ gives $\frac{1}{1+\eta_k\sigma D_k^{-1}}  \approx 1 - \eta_k \sigma D_k^{-1}$ and $\frac{ \eta_k D_k^{-1}}{1+\eta_k\sigma D_k^{-1}} \approx \eta_k D_k^{-1} $. These are exactly the coefficients of AdamD.

---

> > > > > ### Author Response · Authors · 2025-03-19
> > > > > **Authors’ response to Reviewer tSGb**
> > > > >
> > > > > We thank the reviewer for highlighting this interesting interpretation! Sorry for overlooking it earlier. Indeed, our framework does approximate AdamP. We have now added a relevant discussion next to Algorithm1 (AdamD) on Page 16.

---

### Review · Reviewer_Y8UK · 2025-02-25

**Summary Of Contributions:**

The authors consider minimize the objective $ g(x)\coloneqq f(x) + \frac{\sigma^2}{2}\|x\|^2$ over $x\in \mathbb{R}^n$, denoted by (UOP).
Here, $f$ is possibly non-smooth and locally Lipschitz.

The authors consider path-differentiable functions $f$ defined in (Bolte \& Pauwels, 2021). First, note that output of automatic differentiation programs on non-smooth neural network modules like ReLU result in path-differentiable functions. Further, path-differentiable functions admit a conservative field $D_f$ such that for any abs. cont.  mapping $\gamma:[0,\infty]\to \mathbb{R}^n$, $f(\gamma(t)) - f(\gamma(0)) = \int_0^t \max_{d\in D_f(\gamma(s))} \langle \dot \gamma(s), d \rangle ds$.

They consider the Adam-type optimizers to minimize (UOP). Existing works(Xiao et al 2023) that analyze convergence of Adam on path differentiable non-smooth functions require all step-sizes decaying to $0$ to show convergence to stationary points. Additionally, in practice AdamW, where the weight decay ($\sigma$) is decoupled and added only at the update step with step size $\eta_k$, performs much better than Adam. However, the only convergence analysis for AdamW (Zhou et al 2024) requires smoothness and uses an approximate stationarity measure as the convergence metric.


Their main contributions are --
-  **Algorithms**: They propose a larger class of Adam-family of algorithms denoted by (AFMDW), that covers several common adam variants, for instance Adam, Adagrad, Yogi etc that minimizes UOP. Note that AFMDW does not contain AdamW, but these also decouple weight decay and Adam updates like AdamW, with the key difference being that AFMDW applies the preconditioner to the weight decay term as well.
- **Theory**:
    - The proof sketch follows that of (Benaim et al 2005, Davis et al 2020). Here, the interpolated update of AFMDW lies in a perturbed solution of $-D_f$. The perturbation magnitude decreases to $0$ with iterations, so that asymptotically, a stationary point is reached. The magnitude of perturbation depends on noise and step sizes.
    - When the gradient noise is a uniformly bounded martingale difference sequence (Assumptions 3.3(3), 3.15(5)), for decreasing step sizes, the authors show a.s. convergence to stationary points for AFMDW when the step size $\eta_k$ used for updating the iterates is non-diminishing, but the step size $\theta_k$, used to update momentum terms is $o(\frac{1}{\log k})$ (Theorem 3.10).
    - Under a stronger noise assumption and parameterization of $\theta_k$ (Assumption 3.11), the authors show that AFMDW is globally stable, i.e., for every $\epsilon>0$, there exists $\theta_{\max}>0$ with $\lim \sup_{k\to+\infty}\theta_k \leq \theta_{\max}$ such that the distance to stationary points is $\epsilon$ a.s. (Theorem 3.13).
    - For a single time-scale, $\lim_{k\to\infty}\frac{\theta_k}{\eta_k} = \tau_1 $, a specific instance of AFMDW reaches the stationary point of UOP a.s (Theorem 3.22).
- **Experiments**: The authors propose one instantiation of their AFMDW scheme, AdamD, that performs as well as AdamW for several NN tasks involving image classification on ResNets and LSTMs for language modeling.

**References**
- (Bolte \& Pauwels 2021) Conservative set-valued fields, automatic differentiation, stochastic gradient methods and deep learning. Math Programming.
- (Xiao et al 2023) Adam-family Methods for Nonsmooth Optimization with
Convergence Guarantees. JMLR.
- (Zhou et al 2024) Towards Understanding Convergence and Generalization of AdamW. TPAMI.
- (Benaim et al 2005) Stochastic approximations and differential inclusions. SIAM Journal on Control and Optimization.
- (Davis et al 2020) Stochastic subgradient method
converges on tame functions. FoCM.

**Audience:**

Yes

**Broader Impact Concerns:**

None.

**Claims And Evidence:**

Yes

**Requested Changes:**

Please fix the weakneses. The noise assumptions are critical to my recommendation.

Additionally, consider these minor changes.
- **Differences over (Xiao et al 2023):** The current work looks like an extension of (Xiao et al 2023), with a focus on two-timescale analysis of Adam with regularization instead of single-timescale analysis of Adam with gradient clipping. As these papers are very similar in their proof techniques and defining a generic Adam-family, I'd recommend the authors to highlight concrete differences.
- **Possible parallel work (Jin et al 2024)**: This seems to be a parallel work on Adam's convergence guarantees released on arxiv which the authors should consider mentioning in their related works. They consider smooth losses, so the comparison to this paper is direct.
- **Difference between AdamW and AFMDW**: From what I understand, the key reason to introduce AFMDW instead of AdamW is to have the term $\sigma x + m$ which goes to $0$. AdamW instead might have resulted in a weighted version of the same equation with one of the weights depending on $v$ and thus requiring convergence of $v$, as the authors have argued in Page 3 last paragraph. Comparing the surrogate loss function from this paragraph, i.e., $f(x) + \frac{\sigma}{2}\langle x, (\sqrt{v} + \epsilon) \odot x \rangle$, we see that there is a similar term in AFMDW's Lyapunov function in third term of Eq(25), only AFMDW weighs the additional term by $\sigma x + m$. I think the fact that the possible weighted version of $\sigma x$ and $m$ would depend on $v$ and would thus be harder to handle should be discussed to motivate using AFMDW instead of AdamW. Also, this term itself might not be that hard to handle, so the proof of AFMDW could be extended to AdamW.

**References**

- (Xiao et al 2023) Adam-family Methods for Nonsmooth Optimization with
Convergence Guarantees. JMLR.
- (Jin et al 2024) A comprehensive Framework for Analyzing the Convergence of Adam: Bridging the Gap With SGD. arXiv:2410.04458

**Strengths And Weaknesses:**

### Strengths
- **Broad framework**: Their framework AFMDW covers all Adam and Adam-family algorithms except AdamW. Therefore, their convergence results are applicable to all these algorithms.
- **Experimental verification** : Even though their framework does not cover AdamW, their proposed AdamD obtains similar performance to AdamW.
- **Convergence of AFMDW with Non-diminishing $\eta_k$**: Their theoretical results under bounded noise assumptions obtain a.s. convergence to stationary points for non-diminishing $\eta_k$, however with diminising $\theta_k$.
- **Extensive Literature Survey**

### Weaknesses
- **Noise Assumptions**: In Table 1, the authors claim that their methods have WRS (weighted random sampling) and RR (randomized reshuffling) noise assumptions. This is not the case.
    - **WRS** : The noise assumptions corresponding to WRS, Assumptions 3.3 and 3.15, require the noises to be uniformly bounded martingale difference sequence. For a finite-sum objective ($f(x) = \frac{1}{N}\sum_{i=1}^N f_i(x)$) with weighted random sampling (consider weights $\{w_i\}_{i\in [N]}$) with a given batch size $B$, additional assumptions, for instance Lipschitz-ness of $f_i$'s are required for this WRS sampling to satisfy Assumptions 3.3 or 3.15. Assumption 4.1 requires Lipschitz $f_i$'s, so this might imply a certain WRS sampling result as well, however, the authors should either explicitly show that this implies WRS or provide a reference for this. Additionally, the noise assumptions corresponding to WRS should be referred to as bounded martingale difference noise or something equivalent as these are the exact conditions required.
    - **RR**: Assumption 3.11(2) is the additional noise assumption, so I'm considering this as the randomized reshuffling assumption. If $\xi_k$ is the noise at iteration $k$, this assumption requires $\sum_{i=jN}^{(j+1)N-1}\xi_{i+1}=0$ for some $N\in \mathbb{N}$. First of all, this is a very strong assumption as the sum of noise is exactly $0$ on top of being bounded martingale difference. Second, this does not correspond to a random reshuffling arrangement for finite sum objective as the sum of gradient noises in one permutation does not sum to $0$, unless each $f_i(x) = f(x) + c(i)$, where $c$ is a function independent of $x$. I'd recommend the authors to not use the term RR, and provide justification for this noise assumption possibly via examples. Note that this noise assumption is required for global stability of AFMDW.

- **Typos**: Please fix these typos.
    - Assumption 3.11 (2), $\sum_{i=jN}{(j+1)N-1} \xi_{i+1} = 0$ instead of $\sum_{i=jN}{(j+1)N-1} \xi_{k+1} = 0$.
    - Page 13, 2nd paragraph, 2nd line : "can asymptotically find" instead of "can asymptotically finds".
    - Page 26, "Proof of Proposition 3.20" instead of  "Proof ofof Proposition 3.20".

---

> ### Author Response · Authors · 2025-03-18
> **Authors’ response to Reviewer Y8UK**
>
> We thank Y8UK for the valuable comments on the noise assumptions and comparison of related works on Adam! Please find our responses to the questions below. The modifications (marked in blue) and some new references can be found in the revised text.
>
> ### Weakness
> **Noise Assumption：** In Table 1, the authors claim that their methods have WRS (weighted random sampling) and RR (randomized reshuffling) noise assumptions. This is not the case.
>
> ***Ans：***
>
> - **WRS:** We have revised the manuscript accordingly (see Page 10 for details). Consider the setting where $f(x) = \frac{1}{N} \sum_{i = 1}^N f_i(x)$ for locally Lipschitz continuous and path-differentiable functions $\\{f_i\\}$ that admit $\\{\mathcal{D}_{f_i}\\}$ as their conservative fields (i.e., $f$ follows a finite-sum formulation), $\{x_k\}$ is uniformly bounded, and ${g_k}\in {D}\_{f\_{i_k}}(x_k)$ with randomly chosen $i_k \in [N]$ by WRS. Then as demonstrated in (Bolte & Pauwels, 2021; Castera et al., 2021), there exists $\\{\xi_k\\}$ such that $g_k -  \xi\_{k+1} \in \mathcal{D}_f(x_k)$ and $\\{\xi\_{k}\\}$ is a  uniformly bounded martingale difference sequence. Therefore, we can conclude that Assumption 3.3(3) characterizes the evaluation noises when $f$ follows a finite-sum formulation and the indexes are sampled by WRS.
> - **RR:** We have verified that the noise under Randomized Reshuffling (RR) satisfies Assumption 3.11(2). This is because the definition of the perturbed conservative field $\mathcal{D}^{\delta}(z)$ permits perturbations in the argument $z$ (see Definition 2.16), and the gap between iterates within one epoch can be controlled. More detailed discussions are provided after Assumption 3.11 on Pages 12–13.
>
> **Typos:**
>
> ***Ans：*** We have corrected all identified typos in the manuscript.
>
> ### Mirror Changes
> **Differences over (Xiao et al 2023):**
> ***Ans：*** We have added a discussion in the Introduction that highlights the differences between our work and that of Xiao et al. (2023). See Page 3.
>
> **Possible parallel work (Jin et al 2024)：**
>
> ***Ans：*** We have incorporated a comparison with the parallel work by Jin et al. (2024) in the Introduction. See Page 3.
>
> **Difference between AdamW and AFMDW:**
>
> ***Ans：*** We attempted to directly establish the convergence of the original AdamW; however, identifying a suitable Lyapunov function for AdamW remains challenging, especially in nonsmooth nonconvex settings. We believe that exploring the convergence properties of AdamW under these conditions is an interesting and valuable direction for future research. A comment on this issue has been added to the Conclusion. See Page 22.

---

> ### Comment · Reviewer_Y8UK · 2025-04-03
> **Comments on RR**
>
> Thanks for the response, and apologies for not replying during the discussion period.
>
> The derivation of RR shown by the authors is incorrect.
>
> This is because they add up the difference of all the iterates across a single permutation to denote the suboptimality $\delta_k$ at the $k^{th}$ iteration. They have made $\delta_k$ big enough that the evaluation noises can be extremely small and sum up to $0$ in every permutation.
>
> The problem is that this does not satisfy $\lim_{k\to\infty} \delta_k = 0$ (Assumption 3.3(2)) which is required for all results, as only then can the distance to stationary points become arbitrarily small.
>
> Here is a counter example where this does not happen. Consider two critical points $x_1^\star$ and $x_2^\star$ such that their conservative fields contain $0$. Now, consider a simple scenario where a single step/constant number of steps of size $\eta_{min}>0$, the minimum step size in Assumption 3.5, can jump from $x_1^\star$ to $x_2^\star$. Note that this is possible if the conservative fields at these critical points have additional non-zero elements pointing in the correct direction, which can be possible in the non-smooth case. Then, the sequence of iterates after convergence would be $x_1^\star, x_2^\star, x_1^\star, x_2^\star, \ldots$. Note that this sequence only has critical points, so technically the algorithm has converged, but the term $\lvert\lvert  x_1^\star - x_2^\star \rvert\rvert > 0$ and the term $\delta_k$ is a constant multiple of it, so it is also always bounded away from $0$. Instead of using the actual critical points, for any $\epsilon$, we can consider a function such that at any point in the $2\epsilon$ neighborhood of the first critical point $x_1^\star$, the conservative field contains an element pointing towards the second critical point $x_2^\star$ and vice-versa.
>
> I would recommend removing the RR nomenclature from the final version. All other questions have been answered satisfactorily.

---

> > ### Author Response · Authors · 2025-04-03
> > **Authors’ response to Reviewer Y8UK**
> >
> > Thank you for your feedback and for raising this concern. We have revised Theorem 3.3 accordingly to address this concern and marked all the modified contents in red fonts.
> >
> > To establish global stability, the proofs of Lemma 3.12 and Theorem 3.13 do not require Assumption 3.3(2). In particular, while Lemma 3.12 requires Assumption 3.3(3) and Theorem 3.13 requires Assumptions 3.3(1) and (3), we only require $\\{\delta_k\\}$ to be sufficiently small rather than diminishing.  Under RR, This condition is guaranteed if $\\{\eta_k\\}$ is chosen sufficiently small, as demonstrated in the definition of $\\{\delta_k\\}$ in Equation (18). Intuitively, from the uniformly boundedness of the noises $\\{\xi_k\\}$ and Equation (11), sufficiently small stepsizes $\\{\eta_k\\}$ lead to sufficiently small $\\|x_{l+1}-x_l\\|$, hence keeping $\\{\delta_k\\}$ small.
> >
> > We acknowledge that the counterexample does occurs when $\\{\eta_k\\}$ are not sufficiently small. In response, we have updated Theorem 3.13 to include the additional requirement that $\\{\eta_k\\}$ must be sufficiently small. Please refer to Page 13 and the corresponding proof in the Appendix A.

---

> > > ### Comment · Reviewer_Y8UK · 2025-04-03
> > >
> > > Thanks for the quick response. This fixes the counter-example. One detail which I noticed just now, there is no Theorem 3.5 in (Xiao et al. 2023b). I think the authors mean Theorem 3.7 in the current arxiv version. I don't have any further questions.

---

> > > > ### Author Response · Authors · 2025-04-04
> > > > **Authors’ response to Reviewer Y8UK**
> > > >
> > > > Thank you very much for your comments. It should be Theorem 3.7 in (Xiao et al. 2023b). We have corrected that typo in the revised version of our manuscript.

---

### Review · Reviewer_7Xvo · 2025-03-12

**Summary Of Contributions:**

The authors analyze a class of Adam-family methods for quadratically regularized *nonsmooth* nonconvex optimizations. Convergence proofs are provided for path-differentiable functions under reasonable assumptions, extending the literature on convergence results toward the practical implementations of neural networks. From their analysis, an extension of Adam is derived, AdamD. Empirical studies ranging from relatively small to $10^8$ scale models compare Adam, AdamW, and AdamD, which shows the promise of AdamD.

**Audience:**

Yes

**Broader Impact Concerns:**

None is given nor is one needed. This work uses modest computational resources and a wide array of models that would be accessible to anyone implementing optimization algorithms.

**Claims And Evidence:**

Yes

**Requested Changes:**

# Requested Changes
1. Please actually define "UOP". Unconstrained Optimization Problem?
2. Please use consistent algorithmic naming, i.e. renaming framework (1) to (Adam)
3. section 5.1.1: "stepszie" <-- "stepsize"
4. Please provide justification within Section 5.1.1 for using 1/10 stepsizes for AdamD compared to Adam(W).
5. section 5.1.1 p18, "noticessly" <- "notably" ?
6. As a colorblind person I suggest using a dark-to-light colorbar for fig 7 (right now the highest and lowest scores are both dark)
7. I believe it would present nicer to combine panels from Fig 1-4 into a single 4x4 , or at least two 2x4's (also to be consistent with Fig 6 format)
8. [optional] There is an argument that this would be obvious for someone reading past the abstract, but you may consider explicitly introducing "the weight decay term" after (1) to better clarify the last sentence in p3,P5 after (AdamW), and to clarify the "integrated with the function f" statement, and potentially define "coupled" to better clarify what you mean by "decoupled" later on. E.g.
p3,P1,s1: "In the framework (1), the weight decay term, $\sigma x_k$, is integrated with the momentum and estimator terms. This comes from a direct interpretation of differentiating the loss function, where f and the optimization momentum are coupled."
9. Why no AdamW visualization in Figure 7?
10. [optional]: A (more complete?) extension of Fig 6 would be: x-axis is the number of LSTM layers (1:100, say), and y-axis is the final test-set perplexity. I think this would showcase the limits of performance characterization. By the looks of the initial trend, AdamD is doing better for deeper autoregressive nets, which is a great property to showcase. Assuming a standard LSTM implementation, I am guessing they are much smaller (in terms of # parameters) than the models in 5.1.1?

### thoughts (no need to respond)
11. Re: fig 7, it may be interesting to try computing the "area" of good performance as a novel metric. Why no AdamW visualization?
12. It _may_ be worth it to also put AdamW into an algorithm block, right next to Algorithm 1, for direct comparison .
13. Perhaps Section 2.2 could be named (for example) "noise model" instead of probability theory, to contextualize those def'ns

**Strengths And Weaknesses:**

## Strengths
1. I particularly appreciate the clarity of Section 1.1, and the subsequent clarity of a contextualized Section 1.2 (literature review and contribution statements)
2. I am happy with the presentation of the convergence arguments (note: I did not verify each step of the proofs in the appendix)
3. A good array of networks are included in the empirical analysis

## Weaknesses
No significant weaknesses.

---

> ### Author Response · Authors · 2025-03-18
> **Authors’ response to Reviewer 7Xvo**
>
> We thank the reviewer 7Xvo for the valuable comments which have helped to improve the quality of this paper. Below is our detailed response. The modifications (marked in blue) and some new references can be found in the revised text.
>
> ### Regrading the required changes
> **Q1：** Please actually define "UOP". Unconstrained Optimization Problem?
>
> ***A1：*** We have defined "UOP" as the Unconstrained Optimization Problem in the first sentence of Section 1 on Page 1.
>
> **Q2：** Please use consistent algorithmic naming, i.e. renaming framework (1) to (Adam).
>
> ***A2：*** We have revised the algorithm naming accordingly. See Page 2.
>
> **Q3, Q5：** Typos
>
> ***A3, A5：*** The typos have been corrected in Section 5.1.1.
>
> **Q4：** Please provide justification within Section 5.1.1 for using 1/10 stepsizes for AdamD compared to Adam(W).
>
> ***A4：*** We have adjusted the hyperparameter settings for a fairer comparison. Specifically, the learning rate for all three methods is now selected from the same grid: $\eta\in\{5\times10^{-5},10^{-4},5\times10^{-4},10^{-3},5\times10^{-3},10^{-2}, 5\times10^{-2},10^{-1}\}$. More details are provided in Section 5.1.1, 5.1.2 and Apendix B.
>
> **Q7：** I believe it would present nicer to combine panels from Fig 1-4 into a single 4x4 , or at least two 2x4's (also to be consistent with Fig 6 format)
>
> ***A7：*** We have reorganized the figures in Section 5.1.1 as suggested. See Figure 1 on Page 18.
>
> **Q8：** [optional] Explicitly introducing "the weight decay term" after (1) to better clarify the last sentence in p3,P5 after (AdamW), and to clarify the "integrated with the function f" statement, and potentially define "coupled" to better clarify what you mean by "decoupled" later on. E.g. p3,P1,s1
>
> ***A8：***  We have explicitly introduced the weight decay term after (Adam) to clarify the discussion.
>
> **Q9：** Why no AdamW visualization in Figure 7?
>
> ***A9：*** We have added additional experiments to assess the robustness of AdamW with respect to the parameters $\beta$ and $\theta$. These results, presented in Figure 4 on Page 20, show that while AdamD and AdamW exhibit similar overall robustness, the favorable regions for $\theta$ and $\beta$ differ. Specifically,  AdamD is more robust when $\theta$ is small, whereas AdamW performs better when $\theta$ is large.
>
> **Q10：** [optional] Additional experiments regarding more layers of LSTM.
>
> ***A10：*** We have included additional experiments on LSTM models with 4 to 11 layers (see Figure 5 on Page 31). The results confirm that AdamD consistently outperforms Adam and AdamW for models with more than two layers, with the performance disparity becoming increasingly pronounced as the network depth increases. However, for models with more than five layers, overfitting may occur, as indicated by an increase in test perplexity. Due to computational limitations, these experiments were conducted on the relatively small Penn Treebank dataset. We plan to evaluate our algorithm on larger models and datasets in future work.
>
> ### Regrading thoughts
> **Q11：** Re: fig 7, it may be interesting to try computing the "area" of good performance as a novel metric. Why no AdamW visualization?
>
> ***A11：*** We have added the AdamW visualization and computed the percentage of values below 0.6 for each method. See Figure 4 and its caption for further details.
>
> **Q12：** It may be worth it to also put AdamW into an algorithm block, right next to Algorithm 1, for direct comparison.
>
> ***A12：*** We have revised the manuscript accordingly, and an algorithm block for AdamW is now included next to Algorithm 1 (see Page 16).
>
> **Q13：** Perhaps Section 2.2 could be named (for example) "noise model" instead of probability theory, to contextualize those defs
>
> ***A13：*** We have renamed Section 2.2 to "Noise model" (see Page 6) to better reflect its content.

---

### Decision · Action_Editor_MJDs · 2025-04-27

**Recommendation:** Accept as is

**Comment:**

This paper studies the convergence properties of a class of Adam-family methods with decoupled weight decay for minimizing quadratically regularized nonsmooth nonconvex optimization problems. The paper highlights AdamD, a novel version of Adam with weight decay that resembles AdamW (and achieves similar practical performance). However, unlike AdamW, AdamD falls within the studied class, allowing its convergence properties to be derived from the proposed framework.

This paper develops new convergence theory for Adam-type methods in nonsmooth settings. The reviewers noted that including non-asymptotic convergence guarantees could have further strengthened the contribution.

The authors did an excellent job revising the manuscript based on the reviewers' recommendations. All reviewers unanimously recommend acceptance of this work.

**Audience:**

The submission is particularly interesting to the community, as it provides convergence theory for Adam and related methods in a very general setting.

**Claims And Evidence:**

The paper proposes a rigorous framework for analyzing the asymptotic convergence properties of a class of Adam-type methods. The assumptions are clearly stated, and the theoretical statements are carefully derived. The numerical experiments are well described and effectively complement the theoretical findings.